



# Deriving surface soil moisture from reflected GNSS signal observations from a grassland site in southwestern France

Sibo Zhang[1,2], Jean-Christophe Calvet[1], José Darrozes[3], Nicolas Roussel[3], Frédéric Frappart[3,4], and Gilles Bouhours[1]

[1]CNRM, UMR3589 (Meteo-France, CNRS), Toulouse, France

[2]Fondation STAE, Toulouse, France

[3]GET, UMR5563 (CNRS, Université Paul Sabatier, UR254 IRD), Toulouse, France

[4]LEGOS, UMR5566 (CNES, CNRS, IRD, UPS), Toulouse, France

*Correspondence to*: Jean-Christophe Calvet (jean-christophe.calvet@meteo.fr)

**Abstract.** This work aims to assess the estimation of surface volumetric soil moisture (VSM) using the Global Navigation Satellite System Interferometric Reflectometry (GNSS-IR) technique. Year-round observations were acquired from a grassland site in southwestern France using an antenna consecutively placed at two contrasting heights above the ground surface (3.3 or 29.4 m). The VSM retrievals are compared with two independent reference datasets: in situ observations of soil moisture, and numerical simulations of soil moisture and vegetation biomass from the ISBA (Interactions between Soil, Biosphere and Atmosphere) land surface model. Scaled VSM estimates can be retrieved throughout the year removing vegetation effects by the separation of growth and senescence periods and by the filtering of the GNSS-IR observations that are most affected by vegetation. Antenna height has no significant impact on the quality of VSM estimates. Comparisons between the VSM GNSS-IR retrievals and the in situ VSM observations at a depth of 5 cm show a good agreement ($R^2 = 0.86$ and RMSE = 0.04 $m^3$ $m^{-3}$). It is shown that the signal is sensitive to the grass litter water content and that this effect triggers differences between VSM retrievals and in situ VSM observations at depths of 1 cm and 5 cm, especially during light rainfall events.

## 1 Introduction

Soil moisture is a key component in the hydrological cycle and in the soil-plant-atmosphere continuum. It is also important for irrigation management and flood prediction (Rodriguez-Iturbe and Porporato, 2007). However, in situ observations of soil moisture are very sparse and with small sampling volumes. On the other hand, L-band satellite-derived products, for example, from the soil moisture active passive (SMAP) mission or the soil moisture and ocean salinity (SMOS) mission, have a coarse resolution of tens of kilometers (Chan et al., 2016; Kerr et al., 2001). These products consist in surface volumetric soil moisture (VSM) and concern the top soil layer (from the soil surface to a depth of 1 to 5 cm). There is a need to monitor VSM at the local scale in order to validate model simulations, and satellite-derived products. The International Soil Moisture Network (Dorigo et al., 2013) has been collecting such in situ observations. The Committee on Earth



Observation Satellites (CEOS) Land Product Validation group has recommended expanding the soil moisture networks (Morisette et al., 2006). In particular, developing new automatic monitoring techniques to measure VSM is needed.

The Global Navigation Satellite System Interferometric Reflectometry (GNSS-IR) technique has potential to monitor VSM from ground-based (Chew et al., 2014), airborne (Egido et al., 2014; Sánchez et al., 2015; Motte et al, 2016) or spaceborne antennas (Camps et al., 2008). GNSS antennas measure the signal directly emitted by the GNSS satellites, together with the signal reflected by the surface surrounding the antenna. The GNSS-IR technique allows relating the reflected signal to the characteristics of the reflecting surface and to retrieve geophysical variables. Over land, variables such as soil moisture, snow depth and vegetation parameters can be observed using this technique (Larson et al., 2008; Small et al., 2010; Larson and Nievinski, 2013; Wan et al., 2015; Larson, 2016; Roussel et al., 2016; Zhang et al., 2017). GNSS satellites emit active L-band microwave signals (between 1.2 and 1.6 GHz). The L-band signal is less affected by vegetation effects than shorter wavelengths, which is an asset to retrieve surface soil moisture (Kerr et al., 2001). The GNSS-IR footprint can cover up to thousands of square meters, depending on the antenna height and on the satellite elevation angle (Larson et al., 2010; Vey et al., 2016).

In addition to a specially designed antenna to receive the reflected GNSS signal from the land surface (Zavorotny et al., 2014), classical geodetic-quality GNSS antennas can be used to estimate VSM (Larson et al., 2008). Such antennas have an antenna gain pattern optimised for Right Hand Circular Polarization (RHCP) and minimized for Left Hand Circular Polarization (LHCP). A GNSS network called Plate Boundary Observatory (PBO) $H_2O$ with geodetic-quality antennas on ground in western USA is currently used to monitor VSM (Larson et al., 2013; Larson, 2016; Chew et al., 2016) and snow depth (Larson et al., 2009). The basic observation used in this technique is the signal-to-noise ratio (SNR) which is related to temporal changes in the interference between the direct and the reflected GNSS signals. Each Global Positioning System (GPS) satellite repeats the same orbital cycle from one day to another (offset of a few tenths of meter between two adjacent cycles). This property permits monitoring surface changes through time of the environmental conditions surrounding the receiving antenna.

The present day Block II R-M (Replenishment Modernized) and Block II F (Follow-on) GPS satellites now transmit a L2C (1227.60 MHz) civilian signal. Power and precision of the L2C signal are higher than for the L1 C/A signal (1575.42 MHz) transmitted by all GPS satellites. Several previous studies, such as Larson et al. (2008), Larson et al. (2010), Chew et al. (2014), Chew et al. (2016) and Small et al. (2016) exclusively analyzed the SNR data from the GPS L2C signal to retrieve soil moisture. The Block II F satellites also transmit the latest L5 signal (1176.45 MHz) as well, which features even higher power, greater bandwidth and an advanced signal design. There are now seven Block II R-M satellites (Pseudo-Random Noise (PRN) numbers 5, 7, 12, 15, 17, 29 and 31, identifying each satellite) and twelve Block II F satellites (1, 3, 6, 8, 9, 10, 24, 25, 26, 27, 30 and 32).

Due to the motion of the satellites, the direct and reflected signals cause an interference pattern in SNR data. The SNR oscillations depend on known attributes such as the satellite elevation angle, signal wavelength and antenna height. The SNR amplitude and phase can be solved by using the least square estimation (LSE) method (Larson et al., 2008; Chew et al.,



2016). Larson et al. (2008) and Larson et al. (2010) empirically showed that phase correlates with near-surface soil moisture, with values of the coefficient of determination ranging from $R^2 = 0.76$ to $0.90$. This property was used by Chew et al. (2014) to develop an algorithm to estimate surface soil moisture (top 5 cm) for bare soil. They used a physical surface scattering and dielectric permittivity model to derive a relationship between the phase and soil moisture, in volumetric units ($m^3 m^{-3}$). Vey et

al. (2016) validated this algorithm, using field observations acquired during the 2008-2014 period from a site presenting a high percentage of bare soil. They obtained the following $R^2$ and root mean square error (RMSE) scores for VSM retrievals: $R^2 = 0.80$, and RMSE $= 0.05$ $m^3$ $m^{-3}$. However, for vegetated soil the phase of the SNR is also affected by vegetation. Chew et al. (2016) showed that seasonal vegetation effects on phase have to be considered for soil moisture estimation. They also observed that amplitude decreased as vegetation grew. A model database for the SNR from L2C signal was used to remove

most significant vegetation effects. Small et al. (2016) compared different algorithms of GNSS-IR soil moisture retrieval in the presence of vegetation.

Zhang et al. (2017) used the GNSS-IR technique for a wheat field throughout the growth and senescence period, from January to July 2015. They showed that VSM could not be retrieved when the vegetation canopy is too dense, i.e. plant height and simulated dry above-ground biomass larger than one wavelength (~19 cm for L1) and $0.08$ kg $m^{-2}$, respectively.

On the other hand, relative plant height could be retrieved in such conditions.

The objectives of this study are to (1) investigate VSM estimation over a meadow, in contrasting conditions of plant phenology (growth, senescence, after and before cutting), (2) compare the use of L2C and L5 signals, (3) assess the impact of a major change in the height of the receiving antenna above the soil surface, in relation to the SNR sampling interval. Investigating the impact of the sampling interval on VSM retrievals is needed due to the fact that small sampling intervals

(e.g. 1 s) generate a large amount of data (~100 Mb per day for GPS satellites). Larger sampling intervals may be defined to reduce the amount of daily data.

A natural meadow cut once a year is considered, over a long period of more than one year. Past microwave remote sensing studies (e.g. Saleh et al., 2007) have shown that permanent grasslands behave differently from crops. Because permanent grasslands incorporate a litter composed of dead leaves, they can intercept precipitation considerably more than annual

crops. The short growing cycle of annual crops does not allow the accumulation of large amounts of litter material. This property of permanent grasslands can have a major effect on the microwave signal and can perturb the retrieval of VSM, even at L-band (Saleh et al., 2007). Also, the structure of grass canopies differs from the structure of crops such as wheat and this has an impact on the attenuation of the microwave signal by vegetation (Wigneron et al., 2002).

GPS SNR data from both L2C and L5 signals are obtained using a geodetic-quality GNSS antenna. SNR analysis using the

GNSS-IR technique is used to retrieve VSM over a field covered with grass using two approaches: the method proposed by Chew et al. (2016) and the normalization method based on the newly-established scaled wetness index proposed by Zhang et al. (2017). The retrievals from both methods are compared. Another point to underline is the impact of the antenna height (here 2 levels: 3.3 and 29.4 m above the soil surface) on the VSM retrieval. Moreover, the VSM retrievals from two kinds of GPS signal wavelengths (24.45 and 25.40 cm for L2C and L5, respectively) are compared with field observations. We



analyze the vegetation effects on VSM retrieval accuracy. Another important addressed topic is the influence of the sampling interval on the VSM estimates. As the SNR period changes depending on the antenna height, satellite elevation angle, elevation angle change rate and GNSS signal wavelength, the sampling interval has to be adjusted accordingly in order to maintain the VSM retrieval accuracy.

5  Data are described in Section 2 and methods in Section 3. The obtained soil moisture retrievals are presented in Section 4 and compared with independent VSM estimates. Results are discussed in Section 5. And the main conclusions are summarized together with prospects for further research in Section 6.

## 2 Site and data

### 2.1 Site description and validation data

10  The study site is located at the premises of Meteo-France in Toulouse, France, over an experimental field covered with grass (43°34'26"N, 1°22'27"E). Since 2012, this instrumented site includes soil moisture profile observations from the surface down to a depth of 2.2 m. Other measurements such as turbulent fluxes are made in the framework of the Meteopole-Flux project (https://www.umr-cnrm.fr/spip.php?article874&lang=en) and ICOS (Integrated Carbon Observation System, https://icos-eco.fr/). The soil fine earth in the experimental field at a depth of 5 cm consists of 51% sand, 14.5% clay and 15  34.5% silt. The grass was as tall as 30 cm during the experiment time period. The grass was cut twice during the study period. The cutting process took several days and the grass was fully cut on: 7 October 2015 and 9 July 2016, for the 29.4 and 3.3 m antenna observing areas, respectively. Mean in situ VSM observations at 5 and 1 cm depths were performed using precise Delta-T ML2x and low-cost Decagon EC-5 VSM sensors, respectively. Precipitation measurements were made in the experimental field. A small fraction of the precipitation time series was missing. Missing data were replaced by the 20  precipitation data obtained from the SAFRAN atmospheric analysis (Durand et al., 1993, 1999). Additionally, scaled VSM observations at a depth of 1 cm and scaled VSM simulations for the top 1 cm thick soil layer were used as independent benchmarks for validation. VSM simulations were produced using the ISBA (Interactions between Soil, Biosphere, and Atmosphere) land surface model within the SURFEX (version 8.0) modeling platform (Masson et al., 2013). In addition to VSM, simulations included the soil iced water content and the vegetation above-ground dry biomass. The ISBA 25  configuration and the SAFRAN atmospheric analysis used to force the model are described in Lafont et al. (2012).

### 2.2 GNSS data

In this study, GNSS SNR data were acquired using a Leica GR25 multi-constellation and multi-band geodetic receiver equipped with an AR10 antenna during more than one year. Two measurement configurations were explored (Fig. 1). First, from 1 August 2015 to 5 June 2016, the antenna was placed at the top of a building close to the studied grassland, at a height 30  of 29.4 m above the soil surface (43°34'30"N, 1°22'26"E). Second, from 8 June to 6 October 2016, the antenna was moved





on top of a small technical shed located within the meadow, close to the in situ sensors, at a height of 3.3 m above the soil surface. During the first 29.4 m antenna height experiment, the SNR sampling interval was reduced from 10 to 1 s on 19 March. When the antenna height was changed from 29.4 to 3.3 m, the sampling interval remained at a value of 1 s. GNSS SNR data were missing for 24 days: from 1 to 11 January, from 17 to 26 May, and on 1, 6 and 7 June 2016.

In this study, both L2C and L5 SNR data from the GPS Block II R-M and Block II F satellites were used. The ascending and descending parts of the same satellite were processed separately and were considered as independent satellite tracks (Roussel et al., 2015, 2016).

The valid SNR segment for each ascending or descending satellite track was limited based on the available satellite elevation angle range (90° being defined as zenith). For the 3.3 m antenna height, the multipath signature was small at elevation angles

above 30° or below 7°, and the reflecting region (first Fresnel zone, FFZ) often included both ground and surrounding obstructions. Therefore, only data corresponding to elevation angles ranging from 7 to 30° were considered. For a given satellite track, the field observation area was about 300 m², and the observing duration was about one hour (Table 1). The range of instantaneous FFZ areas is indicated in Table 1. After sorting elevation angles, 36 and 21 satellite tracks were available for L2C and L5 SNR data, respectively. The corresponding reflecting points and FFZ areas, obtained using a

reflection location model for GNSS-R (Roussel et al., 2014), are shown in Fig. 1. The successive experimental configurations are listed in Table 2 and shown in Fig. 2.

Measurements from the antenna at a height of 29.4 m were affected by surrounding obstructions (buildings and impervious areas like car park, roads, etc.) and by an under-sampling issue at a sampling interval of 10 s (see Sect. 4.2). In order to cope with these problems, only 6 satellite tracks were used to retrieve VSM from L2C SNR data (GPS PRN 03, 07, 08, 17, 25 and

26), and 4 satellites tracks from L5 SNR data (GPS PRN 03, 08, 25 and 26). Satellite track characteristics and instantaneous FFZ areas are given in Table 1. The selection of satellite tracks and elevation angles was performed by comparing VSM retrievals with the in situ VSM observations described in Sect. 2.1. A larger variety of satellite tracks could be used for the antenna at a height of 3.3 m with 1 s sampling. With a higher antenna, the size of the observed reflecting surface markedly increase (Larson et al., 2010). Although the elevation angle range used for the antenna at 29.4 m is smaller than for the

antenna at 3.3 m (Table 1), a much larger observing area is obtained for each satellite track. More details about the elevation range, the observing time period and approximate observing area for each satellite track are shown in Table 1. The SNR data are typically converted from their native logarithmic units (dB-Hz) to a linear scale (V V⁻¹) (Vey et al., 2016). A low order polynomial curve is fitted to SNR data in order to retain only the multipath interference pattern (Bilich et al., 2008).

## 3 Methods

The modulation of the SNR by the multipath frequency can be expressed as   (Larson et al. 2008,2010, Chew et al. 2016):

$$SNR = A\cos(\frac{4\pi H_0}{\lambda}\sin\theta - \phi) \tag{1}$$



where A is the amplitude of the modulation and $\phi$ *the* phase offset. $\theta$ is the satellite elevation angle, $\lambda$ is the GNSS signal wavelength. $H_0$ is a fixed a priori effective antenna height for each satellite track, which is not known and has to be estimated from the SNR data in snow-free and sparse vegetation conditions (Chew et al., 2016). Based on Eq. (1), SNR phase ($\phi$) can be solved by LSE method, and then this estimated $\phi$ can be used to retrieve VSM.

## 3.1 A new normalized SNR phase method (Zhang et al., 2017)

Normalizing $\phi$ time series ensures compatibility among different satellite tracks (Zhang et al., 2017). Here, $\phi$ is normalized with zero minimum in order to obtain the scaled wetness index ($\phi_{index}$) as the following:

$$\phi_{index} = \frac{\phi - \phi_{\min}}{\phi_{\max} - \phi_{\min}} \qquad (2)$$

where $\phi_{\min}$ and $\phi_{\max}$ are the mean of the lowest and highest 15% of the statistical distribution of $\phi$ for each satellite track during the considered time segment (TS), respectively. This averaging procedure is used in order to filter out outliers corresponding to abnormally high or low $\phi$ estimates. Negative $\phi_{index}$ values are replaced by zero.

Moreover, $\phi_{index}$ can be used to estimate VSM as follows:

$$VSM = \phi_{index} \cdot \left(VSM_{obs\_\max} - VSM_{obs\_\min}\right) + VSM_{obs\_\min} \qquad (3)$$

Similarly to phase computation and to avoid artifacts, $VSM_{obs\_min}$ and $VSM_{obs\_max}$ are the mean of the lowest and highest 15% of daily mean in situ VSM observations at a depth of 5 cm during the considered time segment, respectively. The median VSM estimate from all available satellite tracks is considered as the final VSM estimate per day. In order to better correct for vegetation effects, vegetation growth and senescence were considered as independent time segments instead of applying Eqs. (2-3) to the whole period.

## 3.2 Benchmark zeroed SNR phase method (Chew et al., 2016)

Due to the good linear relationship between $\phi$ and in situ surface VSM, VSM can be estimated for each satellite track (Chew et al., 2016):

$$VSM = S \cdot (\phi - \phi_{\min}) + VSM_{resid} \qquad (4)$$

The $S$ parameter (in $m^3\ m^{-3}$ degree$^{-1}$) is defined using the a priori value, $S = 0.0148\ m^3\ m^{-3}$degree$^{-1}$ from Chew et al. (2016). This value is adapted to situations of low vegetation densinty or cover. In this equation, the $\phi$ time series is zeroed using a minimum phase value ($\phi_{\min}$) for each satellite track. This procedure is useful to ensure compatibility among different satellite tracks. Following Chew et al. (2016), $\phi_{\min}$ is the mean of the lowest 15% of $\phi$ values for each satellite track during the considered time segment. The same condition is used to estimate the $VSM_{resid}$ residual (minimum) soil moisture from the daily mean in situ VSM observations at a depth of 5 cm during the considered time segment. The median VSM estimate from all available satellite tracks during the day is used as the final daily VSM estimate.





### 3.3 Assessment of vegetation effects

SNR amplitude ($A$) is affected by vegetation, which can be used to assess whether or not vegetation effects are significant. Chew et al. (2016) defined the normalized amplitude ($A_{norm}$) as the ratio of amplitude to the average of the top 20% amplitude values. $A_{norm}$ (dimensionless) values below 0.78 indicate that vegetation effects are significant and cannot be
neglected. When vegetation effects are significant, the $S$ parameter value may depart from the value used in Eq. (4). A way to cope with this issue is to apply the Zhang et al. (2017) method for a given time segment presenting consistent vegetation properties. Phase is scaled and $S$ is not needed. The time series in this study is separated into four time segments: (1) TS1, from 1 August 2015 to 18 March 2016 (a vegetation senescence and dormancy period with data acquired from the antenna at 29.4 m using a 10 s sampling interval), (2) TS2, from 19 March to 5 June 2016 (a vegetation growing period with data
acquired from the antenna at 29.4 m using a 1 s sampling interval), (3) TS3, from 8 June to 8 July 2016 (a vegetation growing period with data acquired from the antenna at 3.3 m antenna using a 1 s sampling interval) and (4) TS4, from 9 July to 6 October 2016 (after the grass cutting with data acquired from the antenna at 3.3 m using a 1 s sampling interval).

Another step is to select relevant satellite tracks under significant vegetation effects. This is particularly challenging in dense vegetation conditions. Even in conditions presenting significant vegetation effects, some satellite tracks can be selected to
retrieve VSM. This occurs during TS3, corresponding to low $A_{norm}$ values (Fig. 2). In order to select satellite tracks in such conditions, only tracks presenting a continuity of VSM retrievals with the following vegetation senescence period (TS4) are kept. Only tracks giving similar VSM estimates (difference lower than 0.06 m$^3$ m$^{-3}$) at the end of TS3 and at the beginning of TS4 are used for TS3. This procedure eliminates the tracks corresponding to the most densely vegetated areas in the grass field.

## 4 Results

### 4.1 VSM estimates

Figure 2 presents the VSM estimates derived from both the L2C and L5 SNR data using the normalized SNR phase method (Sect. 3.1) and the vegetation correction method (Sect. 3.3). Results are shown for the whole experiment period from 1 August 2015 to 6 October 2016, and for all the experimental configurations of antenna height, sampling interval, and grass
cutting (time segments).

The first grass cutting event occurs during TS1 but has no effect on $A_{norm}$ because the above-ground biomass is relatively low (less than 0.25 kg m$^{-2}$), as shown in Fig. 2. On the other hand, the second cutting occurring before 9 July 2016 has a significant effect on $A_{norm}$ because, at that time, vegetation is not yet senescent (above-ground biomass is about 0.50 kg m$^{-2}$). Another reason to separate TS3 and TS4 is that mean L2C $A_{norm}$ values are significantly smaller during TS3 (0.56 and 0.94
for TS3 and TS4, respectively).




The scaled wetness indexes ($\phi_{index}$) and VSM estimates are obtained for each of these four time segments. The VSM scores for the four separated time segments are recorded in Table 2. The mean absolute error (MAE), RMSE and $R^2$ scores for senescent, dormant or cut vegetation (TS1 and TS4) are better than during the vegetation growing period (TS2 and TS3). Scatter plot of the in situ VSM observations (N = 409) at a depth of 5 cm versus GNSS VSM retrievals by the Zhang et al.

(2017) method is shown in Fig. 3. The RMSE and the standard deviation of differences (SDD) scores are: RMSE = 0.038 m$^3$ m$^{-3}$ and SDD = 0.035 m$^3$ m$^{-3}$, respectively. The $R^2$ score is equal to 0.86 for merged L2C and L5 SNR data. About the same value is found using only L2C data ($R^2$ = 0.85). The mean bias (0.02 m$^3$ m$^{-3}$) is positive, because the VSM estimates are generally larger than in situ VSM observations at 5 cm depth.

Figure 2 shows that the GNSS VSM retrievals are more sensitive to light rainfall events than in situ VSM observations at 5

cm depth. Such events occur during the summer and autumn of 2016. It can be observed that while GNSS VSM estimates peak at the same time as light rains, the diffusion of water in the soil does not reach the probes at 5 cm depth. This is why the GNSS VSM tends to be larger than in situ VSM. This difference reduces the correlation and increases the errors (Sect. 5.3). In the following sub-sections, more detailed comparisons are presented for antenna heights of 29.4 and 3.3 m.

### 4.2 VSM estimates from a GNSS antenna at 29.4 m above the soil

In most previous studies, VSM was retrieved from GNSS antennas at about 2 or 3 m above the soil surface. Increasing the antenna height can significantly expand the size of the observed areas. In this study, the impact of using a 29.4 m antenna on VSM retrievals is assessed using TS1 and TS2 data. The whole observation area for each track is about 900 m$^2$ or even larger. The grass is cut in TS1, before 7 October 2015. Before grass cutting, the maximum simulated above-ground dry biomass is about 0.25 kg m$^{-2}$ (Fig. 2). For TS1, $A_{norm}$ values are more often than not above 0.78 (Fig. 2). Above this threshold

value, the vegetation effects are not significant (Chew et al., 2016). From mid-August to mid-September (before the start of grass cutting), $A_{norm}$ is slightly smaller than the threshold value, but VSM can be estimated by both methods at these dates. Moreover, no grass cutting effects are observed in the $A_{norm}$ values, which also shows that vegetation effects are not significant. The VSM retrievals from both methods, using the L2C SNR data, are shown in Fig. 4. VSM retrievals from the Zhang et al. (2017) method are closer to the in situ VSM observations at 5 cm depth than retrievals from the benchmark

method. Day to day changes in VSM retrievals are also smaller using the Zhang et al. (2017) method. Figure 5 shows that the calculated linear regression gives a slope of 1.1 for the Zhang et al. retrieval method while for Chew et al. retrievals we obtained a slope of 0.6 which gives an overestimation of the water content close to a factor of 2. Similar results are obtained from the L5 SNR data (Fig. 5). In general, VSM retrievals from the benchmark method tend to be wetter than the in situ observations. In addition to Fig. 5, a comparison of the scores is presented in Table 3. Although $R^2$ scores (> 0.8) from the

benchmark method are similar to those obtained using the Zhang et al. (2017) method, other scores show lower skill levels. For example, RMSE = 0.137 m$^3$ m$^{-3}$ and SDD = 0.068 m$^3$ m$^{-3}$ for L2C data analysed using the benchmark method, against RMSE = 0.042 m$^3$ m$^{-3}$ and SDD = 0.039 m$^3$ m$^{-3}$ using the Zhang et al. (2017) method (Table 3). The regression analysis in Fig. 5 shows that the VSM retrievals by the benchmark method are biased.





Additionally, Fig. 5 and Table 3 show that VSM retrievals using L5 SNR data are close to those derived from L2C SNR data. The retrieval accuracies from L2C and L5 SNR data are similar (Table 3), showing that both L2C and L5 SNR data can be used to retrieve VSM. In Table 2 L2C and L5 SNR data are combined. Results for TS1 in Table 2 show slightly improved scores with respect to those in Table 3. This can be explained by the larger number of available satellite tracks per day.

Overall, the scores obtained during TS1, at a height of 29.4 m and a sampling interval of 10 s are comparable to those obtained in other time segments, including TS2 with a sampling interval of 1 s. The scores (Table 2) in TS2 are similar to the scores in TS1. This does not mean that there is no effect from the sampling interval because vegetation conditions are different in TS1 and TS2. TS2 corresponds to a vegetation growing period. Vegetation growth impacts the reflecting surface and has an impact on the SNR data as illustrated by the fast decrease of $A_{norm}$ values in Fig. 2. Moreover, the SNR data in

TS4 (after grass cutting) are used to assess the impact of changing the sampling interval, without change in vegetation conditions. This is discussed in Section 5.4.

## 4.3 Removing vegetation growth effects from VSM retrievals

Substantial vegetation effects are observed during TS3, at the end of the growing season of 2016. This is evidenced by $A_{norm}$ values lower than 0.78 (Fig. 2). Grass is cut at the end of TS3 (before 9 July 2016). After grass cutting, the SNR $A_{norm}$ values

gradually raise to a relative large value (above 0.78). For example, the daily mean L2C $A_{norm}$ values are 0.67, 0.69, 0.75 and 0.86 from 6 to 9 July 2016, respectively.

In order to remove vegetation effects, the SNR data before and after cutting are considered as distinct datasets (Sects. 3.1 and 3.3). SNR data are used, time segment by time segment, to obtain soil wetness index and then VSM estimates. For L2C (L5), 10 (6) satellite tracks out of 36 (21) are selected for use during TS3. Figure 6a shows the VSM retrievals for each time

segment TS3 and TS4 for L2C SNR data after removing vegetation effects by applying the Zhang et al. (2017) method. The corresponding scores are listed in Table 4. Similar results are obtained for L5 and both L2C and L5 SNR data (Table 4). Results obtained by applying the Zhang et al. (2017) and the Chew et al. (2016) methods to the merged time segments (TS3 and TS4) for L2C SNR data are also shown in Fig. 6b and 6c and in Table 4. SNR-derived VSM are too dry before the cutting and too wet after the cutting (Fig. 6b). After grass cutting, the Chew et al. (2016) method has a good correlation with

*in situ* measurements but gives unrealistic VSM values, larger than 0.5 $m^3m^{-3}$ (Fig. 6c).

## 5 Discussion

### 5.1 Why should growth and senescence be treated separately?

Section 4.3 showed that the VSM retrieval from SNR data during TS3 is of lower quality than during TS4, i.e. after cutting the vegetation. Not all satellite tracks can be used (Table 1) and skill scores are systematically worse (Table 2). Moreover,

Figure 6 shows that a specific calibration (Sect. 3.3) of the retrieval method is needed for TS3. Because both Zhang et al.





(2017) and Chew et al. (2016) methods are based on the minimum phase which is related to the vegetation height and density, the lack of a priori information about this factor is likely to trigger marked discrepancies.

Based on Eq. (1), SNR amplitude $A$ and SNR phase $\phi$ are calculated using the LSE method, assuming that the relative antenna height ($H_0$) for each satellite track is constant across dates and ignoring the impact of the elevation angle change in $A$
(Larson et al., 2008; Larson et al., 2010). The median value of the derived effective antenna height from the SNR data by the Lomb-Scargle periodogram method is considered as the value of the a priori $H_0$ for each satellite track (Chew et al., 2016). This hypothesis is only valid for the dates when the surface is not covered with snow or dense vegetation. Although the real effective antenna height may vary from one day to another, a constant value of $H_0$ is used through time for a given satellite track. This assumption is made in order to ensure the consistency of $\phi$ time series across dates. The a priori $H_0$ value affects
the sinusoid fit, and might cause a systematic bias of $A$ and $\phi$ across dates. When there are significant vegetation effects, the vegetation height affects the effective antenna height (Zhang et al., 2017). This explains why the obtained VSM retrieval time series with merged time segments are not continuous (Fig. 6). Segment by segment normalization is useful to remove such systematic biases and to remove vegetation effects from VSM retrieval. It can be considered as a vegetation correction method.

Figure 7 illustrates the improvement associated to the vegetation correction along with the Zhang et al. (2017) method. The systematic bias caused by the mismatch in $H_0$ is shown. The VSM retrievals do not correlate with the observed VSM ($R^2 = 0.03$). On the other hand, the vegetation correction removes the differences between TS3 and TS4 caused by using the same $H_0$ in both time segments and the VSM retrievals are more consistent ($R^2 = 0.55$).

## 5.2 Are grassland and cropland vegetation effects comparable?

The effect of vegetation on GNSS SNR data is threefold: plant height, above-ground biomass, and litter. At the end of the growing season, plant height and above-ground biomass values can be much larger for annual crops than for grass. On the other hand, while litter is usually missing during the growing phase of annual crops, litter is a characteristic of grasslands (Quested and Eriksson, 2016).

Over our grassland site, the measured grass height at the end of the growing period is 30 cm on 22 June 2016. The grass
height is then only slightly larger than one GNSS wavelength (~ 25 cm for L5). The simulated above-ground biomass by ISBA is shown in Fig. 2. During the summer of 2015, the maximum above-ground biomass slightly exceeds 0.25 kg m$^{-2}$. This short period coincides with $A_{norm}$ values slightly lower than the 0.78 threshold. In June 2016, before the cutting, the above-ground biomass ranges between 0.25 and 0.50 kg m$^{-2}$. The corresponding $A_{norm}$ drops below 0.78, showing that vegetation effects are significant. The simulated green above-ground biomass is 0.39 kg m$^{-2}$ on 22 June 2016, very close to
the observed value of 0.37 kg m$^{-2}$. The litter dry mass is not simulated but a value of 0.29 kg m$^{-2}$ is obtained from in situ observation at the same date, consisting in 0.23 kg m$^{-2}$ of dead leaf material and in 0.06 kg m$^{-2}$ of decomposed leaves. This represents 44 % of the total above-ground organic material.



Zhang et al. (2017) showed that over a wheat field the vegetation gradually replaces the soil as the dominant reflecting surface when plant height becomes comparable to, or larger than, one wavelength, even at relatively low values of the above-ground biomass (an estimate of 0.08 kg m$^{-2}$, is given). In such conditions the $A_{norm}$ drops below 0.78 and the SNR phase is no longer related to soil moisture (Zhang et al., 2017).

This study shows that VSM retrieval above these biomass and plant height thresholds are feasible for grass. However, a limited number of suitable tracks, less affected by vegetation, has to be selected using the grass cutting event (Sect. 3.3). In real practical applications, such tracks are not a priori known and retrieving VSM would be challenging when vegetation effects are significant.

## 5.3 Does the litter affect the GNSS VSM retrieval?

In order to analyze the possible impact of litter on the differences between GNSS VSM and either in situ VSM or ISBA VSM, in situ VSM observations at 5 cm, in situ VSM observations at 1 cm and ISBA VSM simulations at 1 cm are compared with the GNSS VSM retrievals. The GNSS VSM is retrieved applying the Zhang et al. (2017) method to both L2C and L5 SNR data, and the vegetation effects are removed from the retrievals. For ensuring the comparability of these various soil moisture estimates, GNSS retrievals, ISBA 1 cm simulations, in situ 1 cm observations and in situ 5 cm observations are

scaled to dimensionless values.

Figure 8 shows a comparison between the four scaled VSM time series during TS3 and TS4. Soil moisture values tend to increase drastically during precipitation events. Most of the VSM peaks observed in 1 cm in situ observations are also found in 5 cm observations, except for 5-7 July and 5 August 2016. On the other hand, GNSS VSM peaks can occur while in situ VSM observations do not display any response to rain e. g. on 8-14, 25 and 30 June, 30-31 July, and 29 August 2016. A

contrasting result is found comparing GNSS and ISBA VSM estimates, which peak, more often than not, at the same time. As a consequence, the GNSS VSM estimates correlate much better with ISBA VSM ($R^2 = 0.82$) than with in situ VSM observations at 1 cm ($R^2 = 0.63$) and at 5 cm ($R^2 = 0.57$). More scores are presented in Table 5.

The scores resulting from the comparison between scaled VSM validation data and GNSS VSM estimates are separately recorded in Table 6 for each time segment. The highest correlations are with ISBA simulations at 1 cm, for all time

segments. The scores based on in situ VSM observations at 1 cm are similar to those based on in situ VSM observations at 5 cm. For TS4, the correlation with in situ VSM observations at 1 cm is much higher than with those at 5 cm. The main difference between observations at 1 cm and at 5 cm is that the former respond to rainfall events more rapidly. This is illustrated by Fig. 8 for events occurring after 9 July 2016 (TS4). The differences observed between GNSS VSM estimates and in situ VSM observations at 1 cm can be explained by the interception of light rains by the litter. Water contained in the

litter tends to directly reflect the GNSS signal and to prevent the GNSS signal from further penetrating into the soil. This difference is not observed with ISBA simulations because the litter is not implemented in this version of the ISBA model. The good correspondence between ISBA and GNSS VSM estimates can be considered as an artifact: ISBA simulates a VSM peak which does not exist, and the GNSS SNR data are sensitive to a sudden increase in the litter water content and/or to the



rain intercepted by the litter or by the leaves. Another demonstration of the impact of the litter effects can be made, removing rainy days from TS4. The $R^2$ score in Tables 2 and 6 then rises from 0.65 to 0.83.

## 5.4 Does the sampling interval affect the VSM retrieval?

When the antenna height increases, the size of the observing areas is extended. But at the same time the period of the SNR data decreases (Eq. (1)), and a smaller sampling interval is needed to ensure the usability of the SNR data for VSM retrieval. On the other hand, because the SNR period from a high antenna is much smaller, it is possible to use smaller elevation angle ranges and shorter observing time periods per track. The number of complete SNR waveforms is much larger than using a low antenna. We investigate the impact of under-sampling for the 3.3 m antenna and for the 29.4 m antenna.

First, an example of the impact of the sampling interval for the 3.3 m antenna is shown in Fig. 9. L2C SNR observations (N = 90) from GPS PRN 10 ascending tracks during TS4 (after grass cutting) are used to retrieve VSM using various sampling intervals. The Zhang et al. (2017) method is used based on the original 1 s sampling interval and on degraded sampling intervals of 10 and 100 s. During TS4, $A_{norm}$ is above 0.78 (Fig. 2), which also shows that vegetation effects are not significant (Chew et al., 2016). This is a rather dry period but a few rainfall events are observed. They cause changes in the in situ VSM observations at 5 cm, which range between 0.07 and 0.21 $m^3\ m^{-3}$. In Fig. 9, the highest correlation ($R^2 = 0.68$) is for the smallest sampling intervals (1 and 10 s), and the lowest correlation ($R^2 = 0.55$) is observed for the largest sampling interval (100 s). The corresponding statistical scores, resulting from the comparison between in situ VSM observations at a depth of 5 cm and GNSS VSM retrievals are shown in Table 7. As for $R^2$, RMSE and SDD for 1 and 10 s sampling intervals are similar (RMSE = 0.020 $m^3\ m^{-3}$ and SDD = 0.018 $m^3\ m^{-3}$), and denote lower skill for the 100 s sampling interval (RMSE = 0.025 $m^3\ m^{-3}$ and SDD = 0.021 $m^3\ m^{-3}$). Much more day to day variability is observed in the retrievals using a 100 s sampling interval. The impact on the SNR information content of degrading the sampling interval may vary from one day to another. This is illustrated by Fig. 10 for two contiguous days (28 and 29 July 2016). The under-sampling effect at 100 s is more pronounced on 29 July than on 28 July. More pit and peak information is missing on 29 July. This tends to trigger a sharp decrease in the retrieved VSM values. On the other hand, under-sampling tends to increase the retrieved VSM on 28 July. As a result, the retrieved VSM drops by -0.050 $m^3\ m^{-3}$ from 28 to 29 July while the in situ VSM at 5 cm only changes by -0.004 $m^3\ m^{-3}$.

SNR amplitudes are also affected by the sampling interval in TS4. For 29 July 2016, the estimated SNR amplitude is 26 V V$^{-1}$ for both 1 and 10 s sampling intervals, but only 18 V V$^{-1}$ for the 100 s sampling interval. For this example track data acquired by the 3.3 m antenna, the SNR period is about 330 s. There are about 330, 33 and 3 samples in a complete waveform for 1, 10 and 100 s sampling intervals, respectively. Obviously, the 100 s sampling interval does not provide enough samples to retrieve VSM. On the other hand, using a 10 s sampling interval is sufficient for the SNR data acquired by the 3.3 m antenna after cutting the grass.

For the 29.4 m antenna, the sensitivity to the sampling interval is more critical. Fig. 11 shows the SNR oscillations for the GPS PRN 25 ascending track. The SNR period is only about 38 s. With 10 s sampling interval, 3 or 4 samples are available



for a complete waveform. This is about the same situation as for the 100 s sampling interval for the 3.3 m antenna. Figure 11a shows that pit and peak information is missing on 18 March 2016 with respect to the 1 s sampling interval data on the next day in Fig. 11b. Nevertheless, Table 6 shows that the 10 s under-sampling had a limited impact on VSM retrievals during TS1 since the best scores are observed during this segment. This paradoxical result can be explained by the prior use

of the in situ VSM data to select the satellite tracks and the satellite elevation angles (see Sect. 2.2).

## 6 Conclusions

GPS L2C and L5 signal-to-noise-ratio data were obtained at a grassland site in southwestern France during a period of 15 months. A dimensionless scaled wetness index was derived from the SNR observations based on the GNSS-IR technique, using indiscriminately L2C or L5 signals. Surface soil moisture was derived from this scaled wetness index. We show that

accurately estimating soil moisture in units of $m^3$ $m^{-3}$ over such a densely vegetated site is challenging. The method proposed by Chew et al. (2016) is adapted to bare soil or sparse vegetation conditions and cannot be applied as is to our grass site. In order to efficiently limit the impact of perturbing vegetation effects, the grass growth period and the senescence period should be treated separately. While the vegetation biomass effect can be corrected for, the litter water interception influences the observations and cannot be easily accounted for. Overall, a precision of 0.035 $m^3m^{-3}$ is achieved for the whole meadow

growing cycle, and of 0.018 $m^3m^{-3}$ after grass cutting. A suitable sampling interval should be used depending on the antenna height and elevation angle range. Positioning the antenna high up (at 29.4 m in this study) in order to observe a larger area enhances the impact of under-sampling. The signal sampling interval should be better than 10 s in this case. More experiments over contrasting vegetation types are needed to further examine the feasibility of using the GNSS-IR technique.

*Acknowledgments.* The work of Sibo Zhang was supported by the STAE (Sciences et Technologies pour l'Aéronautique et l'Espace) foundation, in the framework of the PRISM (Potentialités de la Réflectométrie GNSS In-Situ et Mobile) project. Authors would also like to thank Eric Moulin and Joel Barrié (CNRM), for their technical support during the field campaign.

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

**Table 1.** Characteristics of the selected satellite tracks from the GNSS antenna at a 29.4 m height and at a 3.3 m height (North is 0° azimuth angle, clockwise rotation).

| Antenna height (m) | Satellite tracks | Elevation angle range (°) | Azimuth angle range (°) | Areas per track (m²) | Instantaneous FFZ area range (m²) | Time duration per track (min) |
|---|---|---|---|---|---|---|
| 29.4 | GPS PRN 03 | 14 to 23 | 216 to 219 | ~ 900 | ~400-150 | 21.6 |
| | GPS PRN 07 | | 168 to 169 | | | 21.2 |
| | GPS PRN 08 | | 166 to 169 | | | 20.3 |
| | GPS PRN 17 | | 223 to 228 | | | 24.0 |
| | GPS PRN 26 | | 168 to 171 | | | 20.3 |
| | GPS PRN 25 | 9 to 17 | 228 to 232 | ~ 2000 | ~1000-300 | 20.7 |
| 3.3 | 36 for L2C (10 during TS3) 21 for L5 (6 during TS3) | 7 to 30 | - | ~ 300 | ~200-10 | ~ 60 |



**Table 2.** Soil moisture scores between daily mean *in situ* VSM observations at a depth of 5 cm and GNSS VSM retrievals (both L2C and L5) from the Zhang et al. (2017) method for the whole experimental period and for four time segments. Best score values among time segments are in bold. MAE is the mean absolute error, RMSE is the root mean square error and SDD is the standard deviation of differences.

| Time segments (TS1 to TS4) | TS1 (from 1 August2015 to 18 March2016) | TS2 (from 19 March to 5 June2016) | TS3 (from 8 June to 8 July2016) | TS4 (from 9 July to 6 October2016) | Whole experiment (from 1 August2015 to 6 October2016) |
|---|---|---|---|---|---|
| Vegetation stages | senescence, after cutting and dormancy | growing | growing | after cutting | all |
| Antenna height (m) | 29.4 | 29.4 | 3.3 | 3.3 | 29.4 or 3.3 |
| Sampling interval (s) | 10 | 1 | 1 | 1 | 10 or 1 |
| N | 220 | 68 | 31 | 90 | 409 |
| Mean bias ($m^3\ m^{-3}$) | 0.016 | 0.028 | 0.023 | **0.006** | 0.016 |
| MAE ($m^3\ m^{-3}$) | 0.031 | 0.039 | 0.035 | **0.013** | 0.029 |
| RMSE ($m^3\ m^{-3}$) | 0.040 | 0.048 | 0.043 | **0.019** | 0.038 |
| SDD ($m^3\ m^{-3}$) | 0.037 | 0.039 | 0.036 | **0.018** | 0.035 |
| $R^2$ | **0.85** | 0.62 | 0.45 | 0.65 | 0.86 |
| *p*-value | 0 | 0 | 0.00001 | 0 | 0 |





**Table 3.** Soil moisture scores between daily mean in situ VSM observations at a depth of 5 cm and GNSS VSM retrievals (either L2C or L5) using either Zhang et al. (2017) or the benchmark Chew et al. (2016) methods during TS1 (SNR data from the 29.4 m antenna with 10 s sampling interval from 1 August 2015 to 18 March 2016). Best score values are in bold. MAE is the mean absolute error, RMSE is the root mean square error and SDD is the standard deviation of differences.

| Time segment (TS1) | In situ 5 cm vs. Zhang et al. (2017) method | In situ 5 cm vs. benchmark method | In situ 5 cm vs. Zhang et al. (2017) method | In situ 5 cm vs. benchmark method |
|---|---|---|---|---|
| Signal | L2C | L2C | L5 | L5 |
| N | 220 | 220 | 220 | 220 |
| Mean bias ($m^3$ $m^{-3}$) | **0.016** | 0.119 | 0.017 | 0.129 |
| MAE ($m^3$ $m^{-3}$) | **0.032** | 0.121 | 0.033 | 0.131 |
| RMSE ($m^3$ $m^{-3}$) | **0.042** | 0.137 | **0.042** | 0.147 |
| SDD ($m^3$ $m^{-3}$) | 0.039 | 0.068 | **0.038** | 0.071 |
| $R^2$ | 0.83 | 0.81 | **0.84** | 0.84 |

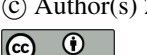



**Table 4.** Soil moisture scores between daily mean *in situ* VSM observations at a depth of 5 cm and GNSS VSM retrievals (either L2C or L5 or both) using either the Zhang et al. (2017) method or the benchmark (Chew et al., 2016) method during TS3 and TS4 (SNR data from the 3.3 m antenna with 1 s sampling interval from 8 June to 6 October 2016). The Zhang et al. (2017) method is used separating time segments. Pooling time segments is shown for comparison with the benchmark method. Best score values are in bold. MAE is the mean absolute error, RMSE is the root mean square error and SDD is the standard deviation of differences.

| Time segments (TS3 and TS4) | In situ vs. Zhang et al. (2017) method | | | | In situ vs. benchmark method |
|---|---|---|---|---|---|
| | separate TS3 and TS4 | | | merged TS3 and TS4 | merged TS3 and TS4 |
| Signal | L2C | L5 | L2C and L5 | L2C | L2C |
| N | 121 | 121 | 121 | 121 | 121 |
| Mean bias (m$^3$ m$^{-3}$) | **0.010** | 0.011 | **0.010** | 0.025 | 0.245 |
| MAE (m$^3$ m$^{-3}$) | 0.019 | **0.018** | **0.018** | 0.044 | 0.248 |
| RMSE (m$^3$ m$^{-3}$) | **0.027** | **0.027** | **0.027** | 0.050 | 0.282 |
| SDD (m$^3$ m$^{-3}$) | 0.026 | **0.025** | **0.025** | 0.044 | 0.140 |
| $R^2$ | 0.55 | **0.60** | 0.57 | 0.03 | 0.03 |





**Table 5.** Soil moisture scores from the comparison between scaled VSM validation data (in situ VSM observations at depths of 1 and 5 cm and ISBA VSM simulations at 1 cm depth) and scaled GNSS VSM retrievals (from both L2C and L5) by the Zhang et al. (2017) method during TS3 and TS4 (SNR data from the 3.3 m antenna with 1 s sampling interval from 8 June to 6 October 2016). Best score values are in bold. MAE is the mean absolute error, RMSE is the root mean square error and SDD is the standard deviation of differences.

| Time segments (TS3 and TS4) | ISBA 1 cm vs. GNSS | In situ 1 cm vs. GNSS | In situ 5 cm vs. GNSS |
|---|---|---|---|
| N | 121 | 121 | 121 |
| MAE | **0.300** | 0.444 | 0.481 |
| RMSE or SDD | **0.435** | 0.637 | 0.699 |
| $R^2$ | **0.82** | 0.63 | 0.57 |





**Table 6.** Soil moisture scores for all time segments (TS1 to TS4) from the comparison between scaled VSM validation data (in situ VSM observations at 1 cm and 5 cm and ISBA VSM simulations at 1 cm) and scaled GNSS VSM retrievals (both L2C and L5) by the Zhang et al. (2017) method. Best score values among time segments are in bold. MAE is the mean absolute error, RMSE is the root mean square error and SDD is the standard deviation of differences.

| Time segments (TS1 to TS4) | TS1 | | | TS2 | | | TS3 | | | TS4 | | |
|---|---|---|---|---|---|---|---|---|---|---|---|---|
| | ISBA 1 cm vs. GNSS | In situ 1 cm vs. GNSS | In situ 5 cm vs. GNSS | ISBA 1 cm vs. GNSS | In situ 1 cm vs. GNSS | In situ 5 cm vs. GNSS | ISBA 1 cm vs. GNSS | In situ 1 cm vs. GNSS | In situ 5 cm vs. GNSS | ISBA 1 cm vs. GNSS | In situ 1 cm vs. GNSS | In situ 5 cm vs. GNSS |
| Antenna height (m) | 29.4 | 29.4 | 29.4 | 29.4 | 29.4 | 29.4 | 3.3 | 3.3 | 3.3 | 3.3 | 3.3 | 3.3 |
| Sampling interval (s) | 10 | 10 | 10 | 1 | 1 | 1 | 1 | 1 | 1 | 1 | 1 | 1 |
| N | 220 | 220 | 220 | 68 | 68 | 68 | 31 | 31 | 31 | 90 | 90 | 90 |
| MAE | 0.32 | 0.33 | **0.30** | 0.47 | 0.58 | 0.56 | 0.34 | 0.54 | 0.65 | 0.33 | 0.33 | 0.38 |
| RMSE or SDD | **0.40** | 0.42 | **0.40** | 0.61 | 0.71 | 0.65 | 0.51 | 0.69 | 0.80 | 0.42 | 0.44 | 0.62 |
| $R^2$ | 0.84 | 0.83 | **0.85** | 0.66 | 0.55 | 0.62 | 0.75 | 0.57 | 0.45 | 0.83 | 0.81 | 0.65 |



**Table 7.** Soil moisture scores from the comparison between daily mean in situ VSM observations at a depth of 5 cm and GNSS VSM retrievals by the Zhang et al. (2017) method during TS4 (after grass cutting, from 9 July to 6 October 2016). The L2C SNR data from GPS PRN 10 ascending tracks were used, which were acquired by the 3.3 m antenna. Best score values are in bold. MAE is the mean absolute error, RMSE is the root mean square error and SDD is the standard deviation of differences

| Time segment (TS4) | 1 s sampling interval | 10 s sampling interval | 100 s sampling interval |
|---|---|---|---|
| N | 90 | 90 | 90 |
| Mean bias | 0.009 | **0.008** | 0.012 |
| MAE ($m^3\ m^{-3}$) | **0.013** | **0.013** | 0.018 |
| SDD ($m^3\ m^{-3}$) | **0.018** | **0.018** | 0.021 |
| RMSE ($m^3\ m^{-3}$) | **0.020** | **0.020** | 0.025 |
| $R^2$ | **0.68** | **0.68** | 0.55 |





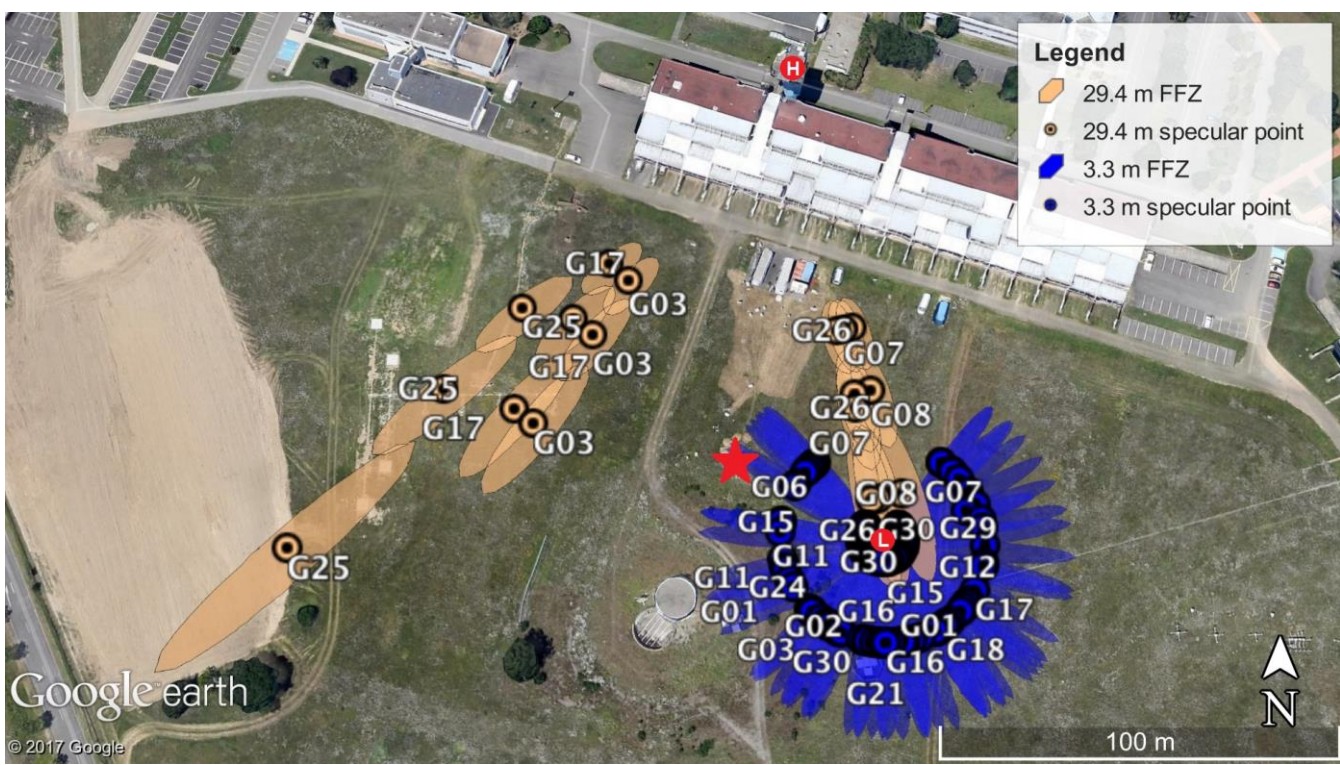

**Figure 1: Experimental site of Meteopole-Flux. The specular reflection points and first Fresnel zone (FFZ) areas from the selected satellite tracks are shown in orange for a 29.4 m GNSS antenna ("H" red dot). The specular reflection points and FFZ areas for a 3.3 m GNSS antenna ("L" red dot) are shown in blue. The red star indicates the location of in situ soil moisture observations. Background geographic information is from Google Earth.**





**Figure 2: Timeline of experiment. (a) Daily GNSS VSM retrieval time series (N = 409) from the Zhang et al. (2017) method using both L2C and L5 SNR data for the whole experimental period (from 1 August 2015 to 6 October 2016) is shown in red line, together with daily mean in situ VSM observations at a depth of 5 cm (green line). The blue line represents the rainfall (daily precipitation in mm day$^{-1}$ can be obtained multiplying by 70). The black lines indicate the grass cutting before 7 October 2015 and before 9 July 2016. The retrievals are obtained separately depending on four time segments (Table 2). (b) The red line represents the above-ground dry biomass (kg m$^{-2}$) of the grass simulated by the ISBA model before grass cutting; and the red dashed line indicates the maximum simulated dry biomass (0.25 kg m$^{-2}$) in 2015. Grass cutting is also shown in black solid lines. The L2C (L5) SNR normalized amplitude ($A_{norm}$, dimensionless) time series is shown in green (blue). Normalization is performed separately for TS1 and TS2, and for the period with data acquired from the 3.3 m antenna using a 1 s sampling interval. The latter corresponds to the merged TS3 and TS4. The black dashed line indicates the $A_{norm}$ threshold (0.78) for evaluating the vegetation effects.**

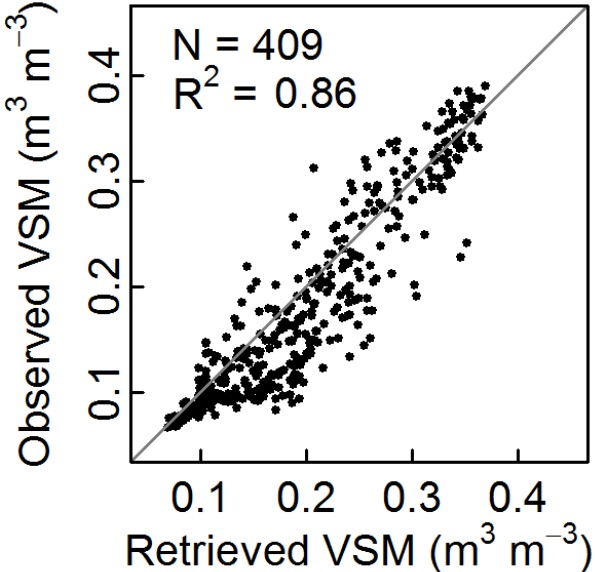

**Figure 3: Scatter plot of daily mean in situ VSM observations (N = 409) at a depth of 5 cm vs. GNSS VSM retrievals (from both L2C and L5) from the Zhang et al. (2017) method for the whole experimental period from 1 August 2015 to 6 October 2016.**





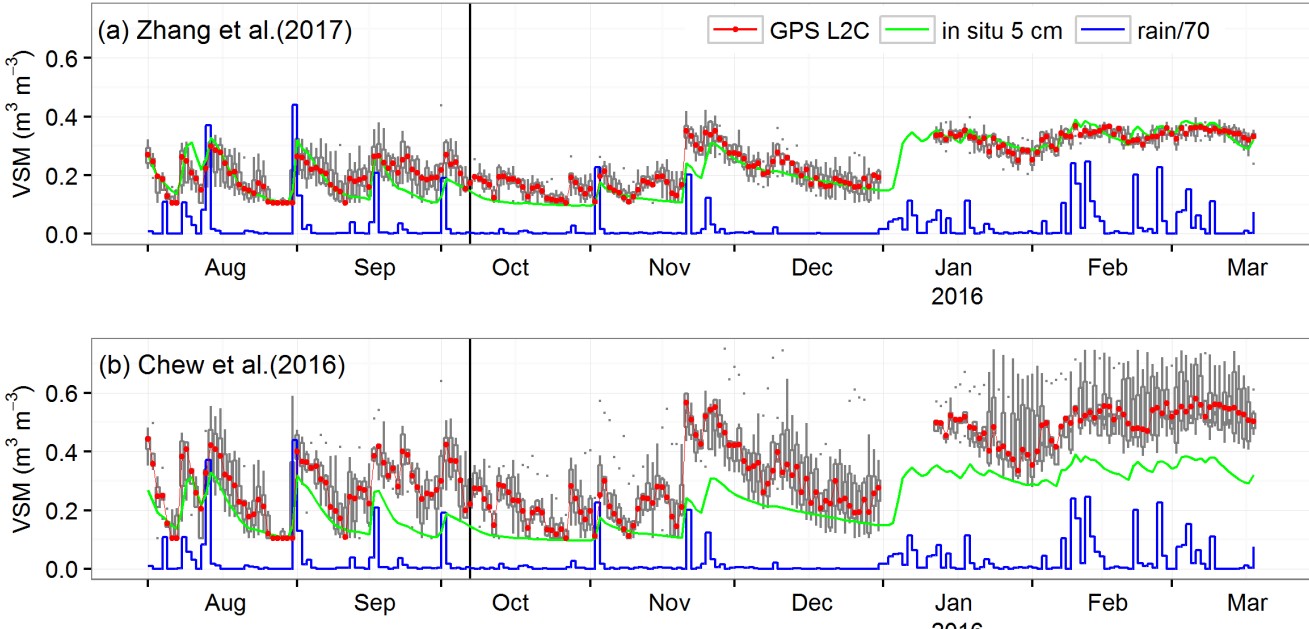

**Figure 4: Median of the daily VSM retrievals (N = 220, red dots) and their daily statistical distribution (grey box plots) for 6 available satellite tracks from: (a) the Zhang et al. (2017) method and (b) the benchmark (Chew et al., 2016) method. Daily mean in situ VSM observations at a depth of 5 cm are shown by the green line. The black line indicates the grass cutting before 7 October 2015. The blue line represents the rainfall (daily precipitation in mm day[-1] can be obtained multiplying by 70). The L2C SNR data acquired by the 29.4 m antenna with a 10 s sampling interval were used to retrieve VSM during TS1 (vegetation senescence and after cutting). Boxes: 25-75% percentiles; bars: maximum (minimum) values below (above) 1.5 IQR (Inter Quartile Range, corresponding to the 25-75%percentile interval); dots: data outside the 1.5 IQR interval.**




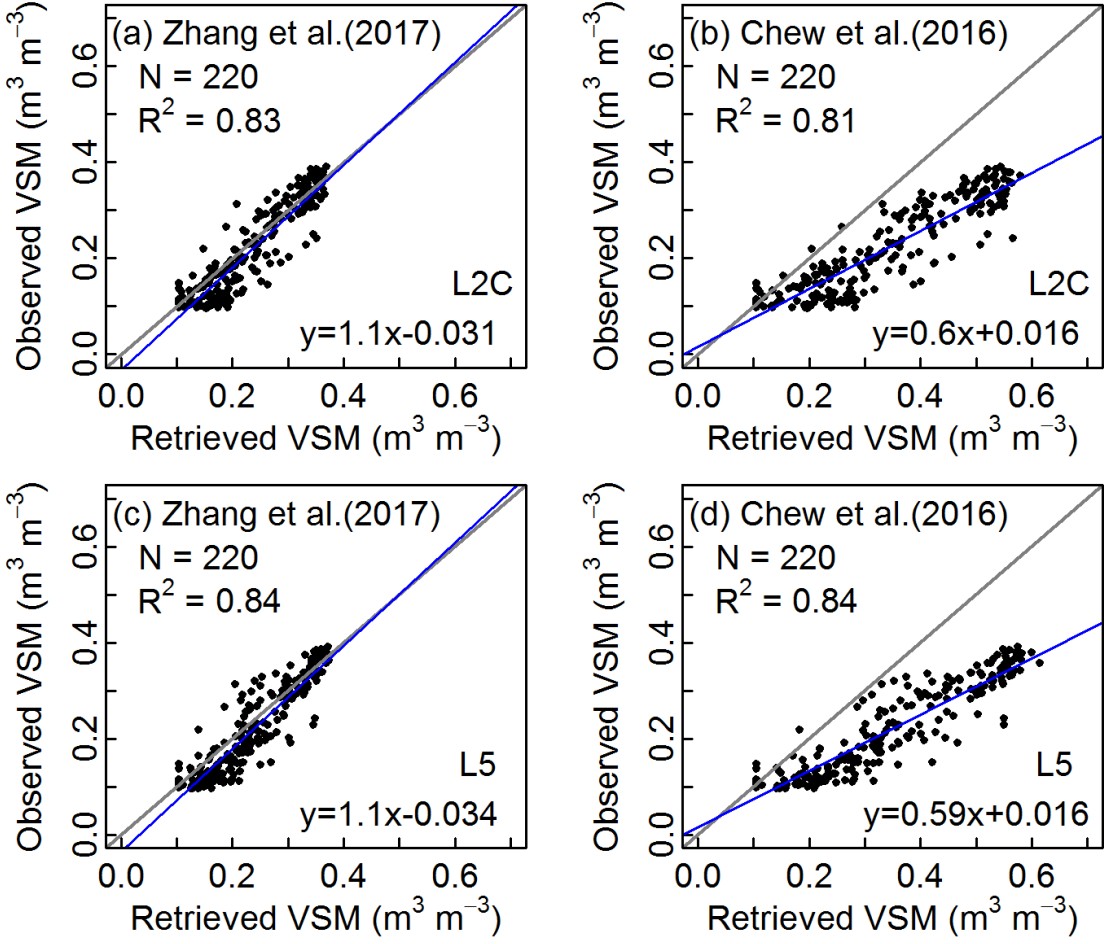

**Figure 5: Scatter plots of daily mean in situ VSM observations at a depth of 5 cm vs. GNSS VSM retrievals (N = 220): from (a, b) L2C SNR data, (c, d) L5 SNR data, using (a, c) the Zhang et al. (2017) method and (b, d) the benchmark (Chew et al., 2016) method. The SNR data acquired by the 29.4 m antenna with a 10 s sampling interval during TS1 were used.**





**Figure 6: Median of the daily VSM retrievals (red lines) from the Zhang et al. (2017) method with (a) separate TS3 and TS4 and removing vegetation effects, (b) merged TS3 and TS4, and from (c) the benchmark (Chew et al., 2016) method with merged TS3 and TS4, using L2C SNR data (from the 3.3 m antenna with 1 s sampling interval) during TS3 and TS4 (from 8 June to 6 October 2016). Daily mean in situ VSM observations at a depth of 5 cm are shown by the green lines. The blue lines represent the rainfall (daily precipitation in mm day$^{-1}$ can be obtained multiplying by 70). The black/orange dashed line indicates the grass cutting before 9 July 2016.**



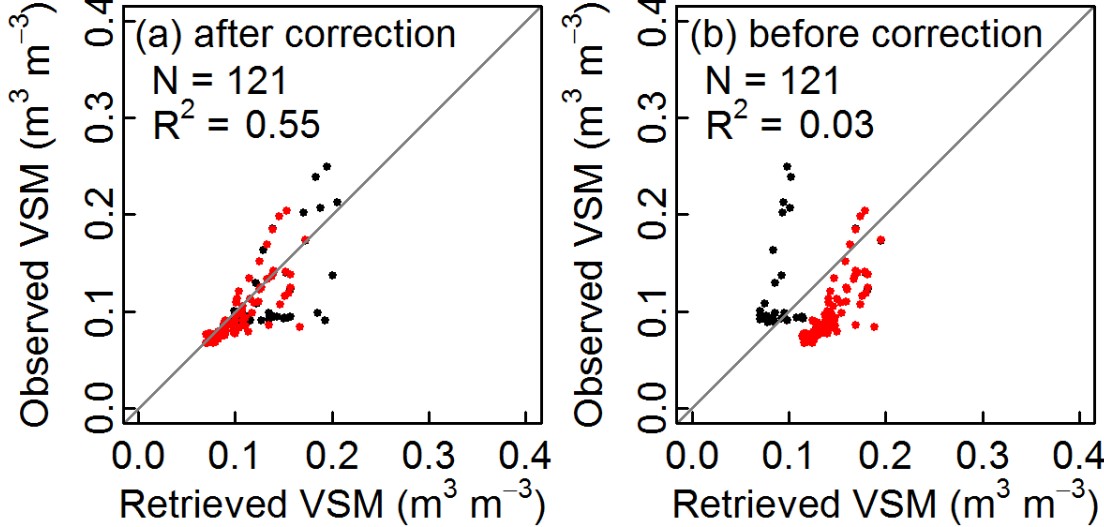

**Figure 7: Scatter plots of daily mean in situ VSM observations (N = 121) at a depth of 5 cm vs. GPS L2C retrievals by the Zhang et al. (2017) method: (a) after vegetation effects correction (with separate TS3 and TS4, corresponding to Fig. 6a) and (b) before correction (with merged TS3 and TS4, corresponding to Fig. 6b). The L2C SNR data acquired by the 3.3 m antenna with 1 s sampling interval were used. Black dots represent the retrievals (N = 31) during TS3; red dots (N = 90) represent the retrievals during TS4 (after grass cutting).**



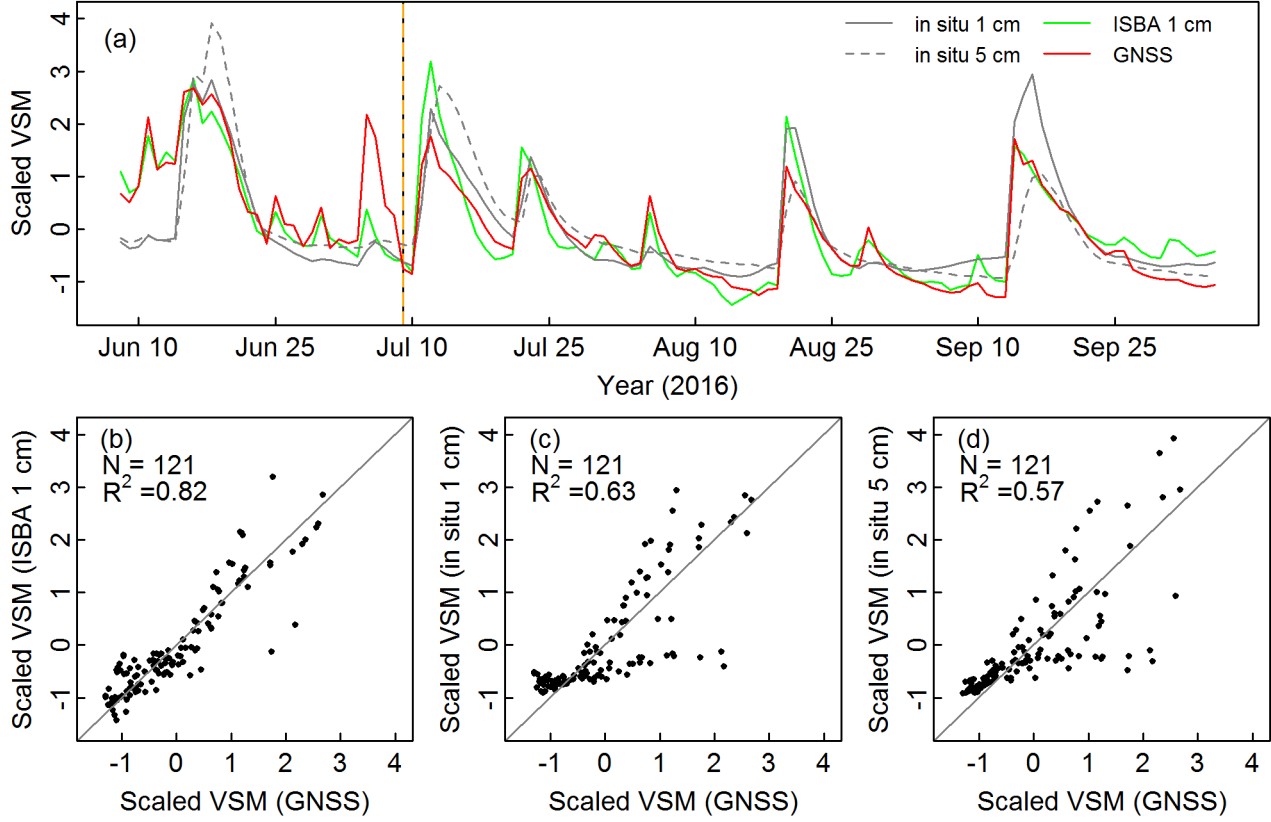

**Figure 8: (a) Scaled GNSS VSM retrieval time series (red line, N = 121) using both L2C and L5 SNR data from the Zhang et al.**
**(2017) method during separate TS3 and TS4, scaled ISBA 1 cm simulations (green line) and scaled in situ VSM observations at 1**
**cm (grey solid line) and at 5 cm (grey dashed line). The SNR data acquired by the 3.3 m antenna with 1 s sampling interval were**
**used during TS3 and TS4. The black/orange dashed line indicates the grass cutting of 9 July 2016. (b, c and d) Scatter plots of**
**scaled ISBA VSM simulations at 1 cm, scaled in situ VSM observations at 1 cm and scaled in situ VSM observations at 5 cm vs.**
**scaled GNSS VSM retrievals, respectively.**



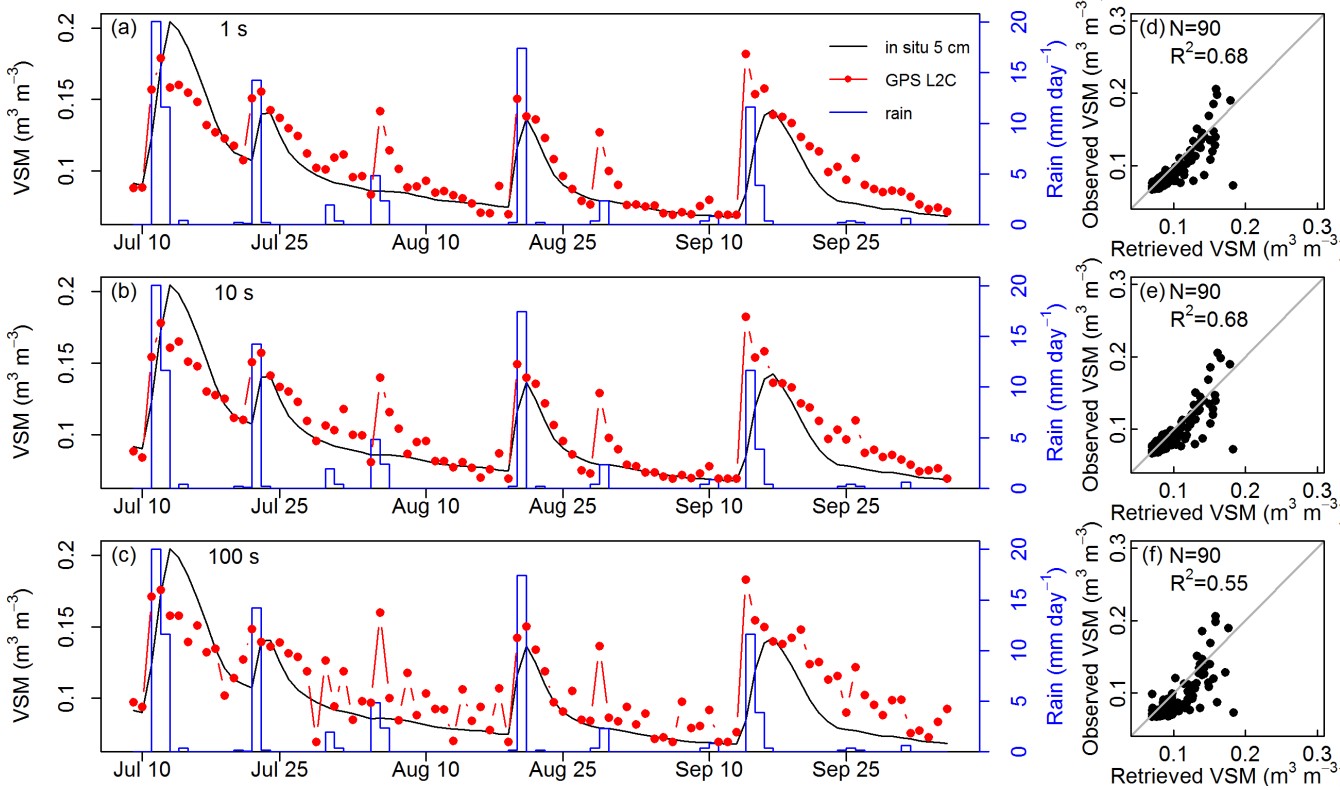

**Figure 9: L2C SNR VSM retrieval time series from the Zhang et al. (2017) method using GPS PRN 10 ascending tracks with different sampling intervals: (a) 1 s, (b) 10 s and (c) 100 s. The L2C SNR data acquired by the 3.3 m antenna during TS4 (after grass cutting) were used. Their corresponding scatter plots are shown in (d), (e) and (f), respectively. Daily mean in situ VSM observations at a depth of 5 cm (black lines) are shown in the left sub-figures, and the blue lines represent the daily precipitation in mm day⁻¹.**


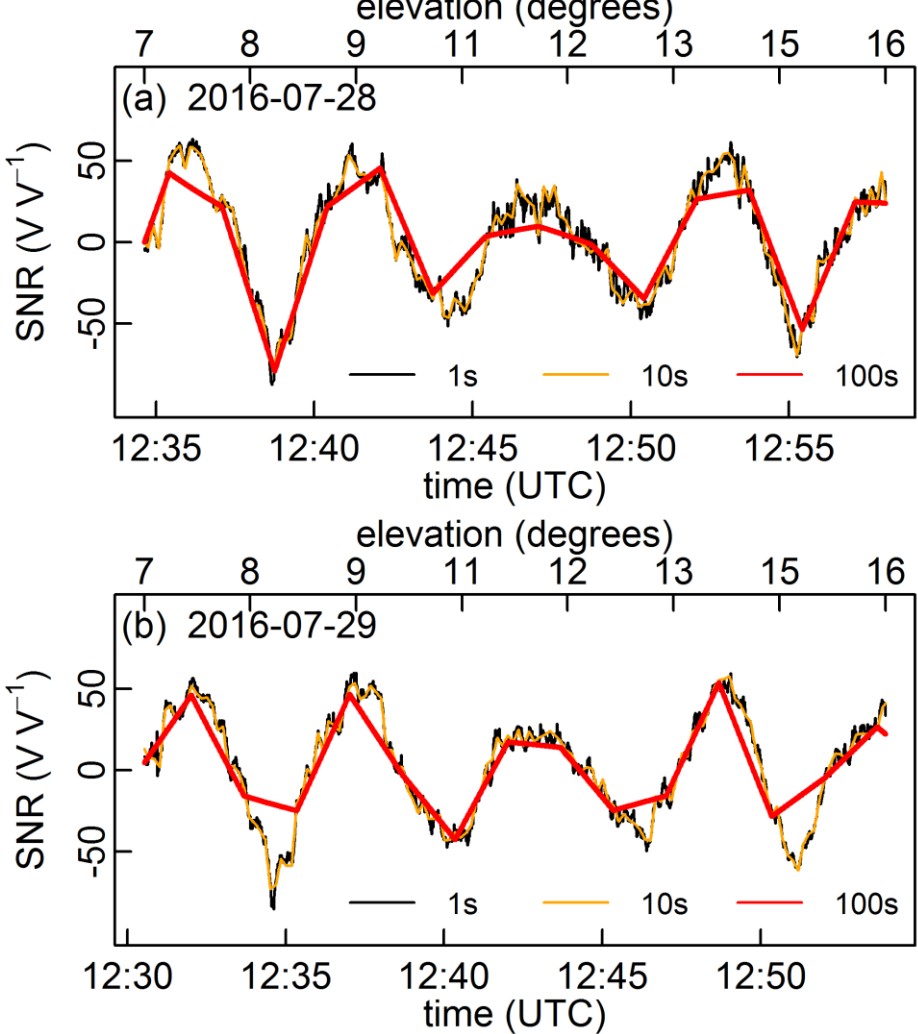

**Figure 10: Two examples of L2C SNR data sets (from the GPS PRN 10 ascending tracks) acquired by the 3.3 m antenna at two contiguous dates: (a) 28 July and (b) 29 July 2016. SNR data with three different sampling intervals at 1, 10 and 100 s are shown in black, orange and red lines, respectively.**





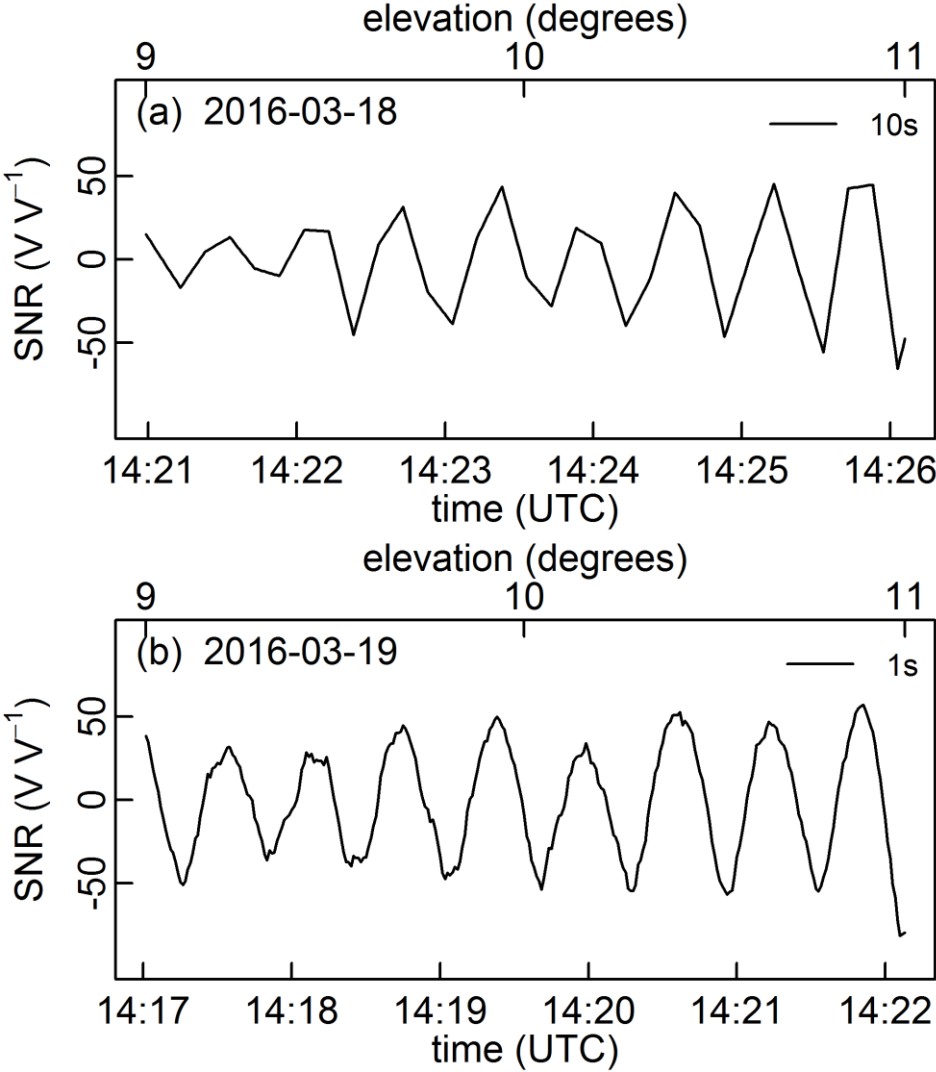

**Figure 11: Two examples of L2C SNR data sets (from the GPS PRN 25 ascending tracks) acquired by the 29.4 m antenna at two contiguous dates: (a) 18 March 2016 (with 10 s sampling interval) and (b) 19 March 2016 (with 1 s sampling interval).**

