# Peer review of "Deriving surface soil moisture from reflected GNSS signal observations from a grassland site in southwestern France"

_Hydrology and Earth System Sciences, 2017_

## Referee Comment (RC1) · Anonymous Referee #1 · 27 Nov 2017

General comments: The authors utilize a geodetic-quality GNSS antenna (AR10 type) in a meadow to test out a soil moisture retrieval algorithm under different stages of natural grass cover growth. They find that their retrieval algorithm performs well and retrieves soil moisture compared to in situ with an RMSE less than 0.04 cm3 cm-3. They compare their results to a 'benchmark' algorithm and find that their algorithm performs better. They also vary the height of the antenna to see if antenna height affects their results, and they also look at the effects from changing the sampling rate. They find that antenna height does not affect their retrievals, but sampling rate does.

Overall, there are two major short comings of this study that must be addressed: *First,
the 'benchmark' algorithm that the authors compare their own retrievals to should NOT be used for this type of antenna. The benchmark algorithm developed in Chew et al. (2016) was created solely for the antennas used in the Plate Boundary Observatory network (Trimble antennas). It is well known that the algorithm would need to be calibrated for use with a different antenna type. The authors should remove the portion of the paper (figures and text) that compare their algorithm to that from Chew et al. (2016). This is a significant portion of the text and dicussion that should be removed, but the paper is still worthwhile without it. *Second, the fact that the authors' retrieval algorithm requires having in situ observations of maximum and minimum soil moisture (Eq. 3) detracts significantly from the usefulness of the algorithm. Of course their algorithm produces soil moisture retrievals within the bounds of the in situ probes–it is effectively scaled by the in situ observations. Furthermore, the authors state that they need min/max in situ observations from both vegetation growth and senescence periods, which then means that they need ancillary vegetation information in order for their algorithm to work. If you need vegetation data and in situ soil moisture probes in order for your algorithm to work, why use GNSS-IR at all? The authors should spend some time re-working their algorithm so that they don't need in situ soil moisture information.

If the authors can address the above two comments, then the paper will be technically correct and will make a more worthwhile contribution to the field of GNSS-IR in general. I know that these are harsh criticisms, and I don't want the authors/editors to think that I don't like the paper–overall, I enjoyed reading it. It is well organized and clearly written. I think reporting their retrieval results is worthwhile, and removing the comparison with the benchmark algorithm will not detract from the paper.

Specific comments: Page 2, line 5: You should make it clear that GNSS-IR is not used for spaceborne applications, as you reference in the Camps et al. (2008) paper. The spaceborne technique is very different from GNSS-IR.

Page 8, line 9: Isn't another way of saying this, is that the sensing depth of GNSS-IR is less than 5 cm? This has been found in previous studies for GNSS-IR (Chew et al.,

2014) and for L-band microwave remote sensing in general (Shellito et al., 2016, GRL).

The comparison with the land surface model is a bit rushed and perhaps not needed. As you know, there are a variety of different land surface models, each with their own parameterizations of the land surface. There aren't enough details provided about the land surface model for readers to understand its advantages and shortcomings. Was it parameterized for this particular field? What is the spatial resolution of the model? The authors do not spend much time with comparing their results to the model output, so it would be easy to remove this part of the paper.

With regards to the sampling rate discussion–are you not just exploring effects of sampling lower than the required Nyquist sampling frequency for a given antenna height?

Technical corrections: Figure 2 needs a second y-axis for Anorm. I understand they are scaled between 0-1 just like you have your biomass values, but it's a bit confusing without an extra label.

---

## Referee Comment (RC2) · Anonymous Referee #2 · 27 Nov 2017

**OVERVIEW**

The manuscript investigates the use of the Global Navigation Satellite System Interferometric Reflectometry (GNSS-IR) technique for soil moisture retrieval. Specifically, one year of observations were acquired at a grassland site in France by using an antenna at 2 different heights (3.3 and 29.4 m). GNSS-IR data are compared with ground-based reference measurements and the effect of vegetation, litter water interception, sampling

interval and antenna height is analysed on the accuracy of the measurements.

**GENERAL COMMENTS**

GNSS-IR represents a new approach for measuring soil moisture and surely deserves to be investigated. Specifically, the potential of using GNSS-IR measurements for monitoring soil moisture over large areas might represent an important step forward in our capability of measuring soil moisture at field scale. The manuscript is well written and clear and, hence, I have no major comments to be addressed. I believe the paper might be published after considering the minor comment I reported below.

1) The same authors (nearly) published a paper in 2017 with a very similar purpose. I can see the differences between the two papers, and hence I believe this paper should be published. However, I strongly suggest to clearly underline the differences between the two papers and the main innovative aspects (e.g., antenna height, analysis of vegetation effect) of the current study.

2) In the description of the study area, more details should be provided. At which depth are installed the surface measurements? How many rain gauges are available in the study area? How many soil moisture stations (I guess one)? Which model is used for simulating soil moisture? I suggest adding all these details in the revised manuscript.

3) At Page 5, lines 17-20 it reads that only some satellite tracks are selected based on the comparison with in situ measurements. I was wondering how the authors will select the tracks if in situ soil moisture observations are not available, it should be clarified.

4) At Page 5, line 28, what is the "multipath interference pattern". Please clarify.

On this basis, I believe the paper deserves to be published only after a minor revision.

---

## Author Comment (AC1) · 24 Jan 2018

The authors thank anonymous reviewer 1 for his/her review of the manuscript and for the fruitful comments.

1.1 [General comments: The authors utilize a geodetic-quality GNSS antenna (AR10 type) in a meadow to test out a soil moisture retrieval algorithm under different stages of natural grass cover growth. They find that their retrieval algorithm performs well and retrieves soil moisture compared to in situ with an RMSE less than 0.04 cm3 cm-3. They compare their results to a 'benchmark' algorithm and find that their algorithm performs better. They also vary the height of the antenna to see if antenna height afprinter_link

fects their results, and they also look at the effects from changing the sampling rate. They find that antenna height does not affect their retrievals, but sampling rate does. Overall, there are two major short comings of this study that must be addressed: *First, the 'benchmark' algorithm that the authors compare their own retrievals to should NOT be used for this type of antenna. The benchmark algorithm developed in Chew et al. (2016) was created solely for the antennas used in the Plate Boundary Observatory network (Trimble antennas). It is well known that the algorithm would need to be calibrated for use with a different antenna type. The authors should remove the portion of the paper (figures and text) that compare their algorithm to that from Chew et al. (2016). This is a significant portion of the text and discussion that should be removed, but the paper is still worthwhile without it.]

Response 1.1:

Yes. We agree with Reviewer 1. Different GNSS receiving antennas and also various ground situations could affect the a priori S value used in the Chew et al. (2016) algorithm. We will remove the Chew et al. (2016) results from Figs. 4, 5 and 6 and from Tables 3 and 4, while noting that results very similar to those presented in the revised Fig. 5 can be obtained by multiplying by 0.6 the S value used by Chew et al. (2016).

1.2 [*Second, the fact that the authors' retrieval algorithm requires having in situ observations of maximum and minimum soil moisture (Eq. 3) detracts significantly from the usefulness of the algorithm. Of course their algorithm produces soil moisture retrievals within the bounds of the in situ probes–it is effectively scaled by the in situ observations. Furthermore, the authors state that they need min/max in situ observations from both vegetation growth and senescence periods, which then means that they need ancillary vegetation information in order for their algorithm to work. If you need vegetation data and in situ soil moisture probes in order for your algorithm to work, why use GNSS-IR at all? The authors should spend some time re-working their algorithm so that they don't need in situ soil moisture information. If the authors can

address the above two comments, then the paper will be technically correct and will make a more worthwhile contribution to the field of GNSS-IR in general. I know that these are harsh criticisms, and I don't want the authors/editors to think that I don't like the paper–overall, I enjoyed reading it. It is well organized and clearly written. I think reporting their retrieval results is worthwhile, and removing the comparison with the benchmark algorithm will not detract from the paper.]

Response 1.2:

Yes. Figure 7 clearly shows that using GNSS-IR to retrieve VSM values in m3 m-3 when significant changes in vegetation effects occur is challenging. The need to harmonize VSM retrievals from time segments 3 and 4 is related to the cutting of the grass when vegetation effects are pronounced (Anorm is lower than 0.78, see Fig. 1). This rather negative result is, still, technically correct. In the revised version of the paper, we will better emphasize that monitoring VSM using a GNSS network is difficult when vegetation effects are noticeable. However, we show that one may use the information from Anorm data to define time segments when scaled VSM time series can be used: grass cutting can be detected from the rapid rise in Anorm value. This is an encouraging result. In this study, we used independent VSM in situ observations to harmonize the VSM time series across time segments 3 and 4. Since in situ observations are not extensively available, this technique is not readily applicable at other sites. In practice, one could possibly use a data assimilation framework able to integrate the VSM retrievals into model VSM simulations such as those produced by the ISBA land surface model (Albergel et al., 2017). In such Land Data Assimilation Systems (LDAS), a complex seasonal rescaling of VSM observations is needed when the observations are not properly decontaminated from vegetation effects (Stoffelen et al., 2017). LDAS are usually used for satellite observations but can also integrate ground observations. Proposing a complete protocol to apply this method to local GNSS antennas is out of the scope of this work. Observations at a large number of sites would be needed. It can be concluded that more research is needed to use GNSS-IR in densely vegetated

areas. These considerations will be included in the Discussion and in the Conclusion Sections.

New references:

Albergel, C., S. Munier, D. J. Leroux, H. Dewaele, D. Fairbairn, A. L. Barbu, E. Gelati, W. Dorigo, S. Faroux, C. Meurey, P. Le Moigne, B. Decharme, J.-F. Mahfouf, J.-C. Calvet: Sequential assimilation of satellite-derived vegetation and soil moisture products using SURFEX_v8.0: LDAS-Monde assessment over the Euro-Mediterranean area, Geosci. Model Dev., Geosci. Model Dev., 10, 3889–3912, https://doi.org/10.5194/gmd-10-3889-2017, 2017.

Stoffelen, A., S. Aaboe, J.-C. Calvet, J. Cotton, G. De Chiara, J. Figua-Saldana, A. A. Mouche, M. Portabella, K. Scipal, W. Wagner: Scientific developments and the EPS-SG scatterometer, IEEE J. Sel. Topics Appl. Earth Obs. Remote Sens., 10 (5), 2086-2097, https://doi.org/10.1109/JSTARS.2017.2696424, 2017.

1.3 [Specific comments: Page 2, line 5: You should make it clear that GNSS-IR is not used for spaceborne applications, as you reference in the Camps et al. (2008) paper. The spaceborne technique is very different from GNSS-IR.]

Response 1.3:

Yes, this is confusing, we agree. We will delete this sentence.

1.4 [Page 8, line 9: Isn't another way of saying this, is that the sensing depth of GNSS-IR is less than 5 cm? This has been found in previous studies for GNSS-IR (Chew et al., 2014) and for L-band microwave remote sensing in general (Shellito et al., 2016, GRL). The comparison with the land surface model is a bit rushed and perhaps not needed. As you know, there are a variety of different land surface models, each with their own parameterizations of the land surface. There aren't enough details provided about the land surface model for readers to understand its advantages and shortcomings. Was it parameterized for this particular field? What is the spatial resolution of the model?

[Figure]

The authors do not spend much time with comparing their results to the model output, so it would be easy to remove this part of the paper.]

Response 1.4:

We agree that more details about the model simulations need to be provided, in particular on the soil modeling part. This will be done in the revised version of the manuscript. The model version we use has been designed for generic country-scale simulations over France at a spatial resolution of 8 km x 8 km. Sub-grid vegetation types are represented and soil moisture and soil temperature profiles are simulated for each vegetation type, independently of other vegetation types. In this study, the C3 grassland plant functioning type and a multilayer representation of the soil hydrology are considered. The model soil depth is 12 m, with 15 layers and the layer thickness increases from the top surface layer to the deepest layers (Decharme et al., 2011). It must be noted that the SAFRAN precipitation forcing is based on ground observations and is quite realistic (Quintana-Segui et al., 2008). Model simulations are useful to assess the litter interception effect (Fig. 8). We also agree that the sensing depth of GNSS-IR is not sufficiently discussed in Section 5.3. We will include the suggested references accordingly (in particular Shellito et al., 2016). We will make clear that the better agreement with scaled model VSM simulations is probably due to compensating errors triggered by the lack of representation in the model of a litter layer above the soil surface.

New references:

Decharme, B., Boone, A., Delire, C., and Noilhan, J.: Local evaluation of the Interaction between Soil Biosphere Atmosphere soil multilayer diffusion scheme using four pedotransfer functions, J. Geophys. Res., 116, D20126, https://doi.org/10.1029/2011JD016002, 2011.

Quintana-Segui, P., Lemoigne, P., Durand, Y., Martin, E., Habets, F., Baillon, M., Canellas, C., Franchisteguy, L., and Morel, S.: Analysis of near surface atmospheric variables: Validation of the SAFRAN analysis over France, J. Appl. Meteorol. Clim., 47,

92–107, https://doi.org/10.1175/2007JAMC1636.1, 2008.

Shellito, P. J., Small, E. E., Colliander, A., Bindlish, R., Cosh, M. H., Berg, A. A., Bosch, D. D., Caldwell, T. G., Goodrich, D. C., McNairn, H., Prueger, J. H., Starks, P. J., van der Velde, R., and Walker, J. P.: SMAP soil moisture drying more rapid than observed in situ following rainfall events, Geophys. Res. Lett., 43, 8068–8075, https://doi.org/10.1002/2016GL069946, 2016.

1.5 [With regards to the sampling rate discussion–are you not just exploring effects of sampling lower than the required Nyquist sampling frequency for a given antenna height?]

Response 1.5:

No. In the examples shown in Figs. 9, 10 and 11, the SNR frequency is lower than the Nyquist frequency (half of the sampling frequency). This will be indicated in the revised version of the paper.

1.6 [Technical corrections: Figure 2 needs a second y-axis for Anorm. I understand they are scaled between 0-1 just like you have your biomass values, but it's a bit confusing without an extra label. ]

Response 1.6:

Yes. We will add a second y-axis for Anorm.

---

## Author Comment (AC2) · 24 Jan 2018

The authors thank anonymous reviewer 2 for his/her review of the manuscript and for the fruitful comments.

2.1 [OVERVIEW The manuscript investigates the use of the Global Navigation Satellite System Interferometric Reflectometry (GNSS-IR) technique for soil moisture retrieval. Specifically, one year of observations were acquired at a grassland site in France by using an antenna at 2 different heights (3.3 and 29.4 m). GNSS-IR data are compared with ground-based reference measurements and the effect of vegetation, litter water interception, sampling interval and antenna height is analysed on the accuracy of the measurements. GENERALCOMMENTS GNSS-IR represents a new approach for measuring soil moisture and surely deserves to be investigated. Specifi-cally, the potential of using GNSS-IR measurements for monitoring soil moisture over large areas might represent an important step forward in our capability of measuring soil moisture at field scale. The manuscript is well written and clear and, hence, I have no major comments to be addressed. I believe the paper might be published after considering the minor comment I reported below. 1) The same authors (nearly) published a paper in 2017 with a very similar purpose. I can see the differences between the two papers, and hence I believe this paper should be published. However, I strongly suggest to clearly underline the differences between the two papers and the main innovative aspects (e.g., antenna height, analysis of vegetation effect) of the current study.]

Response 2.1:

Yes. We will better describe the differences between the two studies throughout the manuscript.

In this study, both L2C and L5 signals were observed over a meadow during a rather long period of time of about 15 months using an AR10 antenna at contrasting heights (3.3 or 29.4 m) above the soil surface, while in Zhang et al. (2017) the L1 C/A signal was observed over a wheat field during a shorter period of about 7 months using an AS10 antenna at a constant height of 2.5 m above the soil surface.

A key difference between the two papers is related to the observed vegetation canopy. The studied grass and wheat (this study and Zhang et al. 2017, respectively) presented contrasting characteristics. While considered multi-species permanent grassland incorporated a litter composed of dead leaves, the wheat crop consisted of a single plant species with no litter. Another difference between the two canopies is that maximum height of the grass (about 0.3 m) was much lower than maximum height of the wheat (about 1 m). A large difference could also be noticed in maximum above-ground dry

biomass values: less than 0.5 kg m-2 for grass, about 1 kg m-2 for wheat.

While VSM could not be retrieved by Zhang et al. (2017) after wheat tillering, i.e. for plant height larger than 0.2 m, we could retrieve scaled VSM values throughout time segments of the grass growing and senescence phases. However, retrieving VSM values in m3 m-3 was challenging and required a seasonal rescaling to account for vegetation effects (see Fig. 7).

2.2 [2] In the description of the study area, more details should be provided. At which depth are installed the surface measurements? How many rain gauges are available in the study area? How many soil moisture stations (I guess one)? Which model is used for simulating soil moisture? I suggest adding all these details in the revised manuscript.]

Response 2.2:

Three ThetaProbes measured VSM at a depth of 5 cm and were located within a few meters of each other (red star in Fig. 1). The mean value was derived from these three probes to represent the in situ VSM observations at 5 cm. One EC-5 Decagon VSM sensor was installed at 1 cm to measure VSM at 1 cm.

One rain gauge was available close to the in situ soil moisture sensors.

Moreover, VSM simulations for the top 1 cm were produced using the ISBA (Interactions between Soil, Biosphere, and Atmosphere) land surface model within the SURFEX (version 8.0) modeling platform (Masson et al., 2013). The ISBA model used the atmospheric forcing data produced by the SAFRAN atmospheric analysis of Météo-France. We used a model configuration corresponding to the C3 grassland plant functioning type and a multilayer representation of the soil hydrology. The model soil depth was 12 m, with 15 layers and the layer thickness increased from the top surface layer to the deepest layers (Decharme et al., 2011).

New reference:

Decharme, B., Boone, A., Delire, C., and Noilhan, J.: Local evaluation of the Interaction between Soil Biosphere Atmosphere soil multilayer diffusion scheme using four pedotransfer functions, J. Geophys. Res., 116, D20126, https://doi.org/10.1029/2011JD016002, 2011

2.3 [3] At Page 5, lines 17-20 it reads that only some satellite tracks are selected based on the comparison with in situ measurements. I was wondering how the authors will select the tracks if in situ soil moisture observations are not available, it should be clarified.]

Response 2.3:

Yes. This is a clear limitation of using high antenna heights. It must be noted that this limitation only affected measurements at a height of 29.4 m and was caused by the more complex experimental constraints in this configuration (e.g. possible parasitic signal reflection on buildings). For the low antenna configuration (3.3 m), this additional data sorting was not needed and all available satellite tracks with a complete elevation angle range (between 7 and 30°) were used.

We will clarify this in the revised manuscript.

2.4 [4] At Page 5, line 28, what is the "multipath interference pattern". Please clarify.]

Response 2.4:

For a static receiver, the SNR is governed to a large extent by the interference pattern (IP). The IP is defined as the coherent summation of direct and reflected GNSS signals on the in-phase and quadrature space (Zavorotny et al., 2014). This coherent summation generates an IP where high and intermediate frequencies distinct from noise frequencies, are related to the difference of travelled distance between direct and reflected waves. The IP can be characterized with GNSS receivers using either (1) two antennas (e.g. Rodriguez-Alvarez et al., 2011) or (2) one antenna (e.g. Larson et al., 2008; Chew et al., 2014; Zhang et al., 2017). In this study we used the one-antenna

IP technique as illustrated by Fig. 1 in Larson et al. (2016) for a simple planar and horizontal ground reflection. We will clarify this in Section 2.

New reference:

Rodriguez-Alvarez, N., Vall-llossera, M., Camps, A., Bosch-Lluis, X., Monerris, A., Ramos-Perez, I., Valencia, E., Marchan-Hernandez, J.F., Martinez-Fernandez, J., Baroncini-Turricchia, G., Perez-Gutierrez, C., and Sanchez, N.: Land geophysical parameters retrieval using the interference pattern GNSS-R technique, IEEE Trans. Geosci. Remote Sens., 49, 71–84, https://doi.org/10.1109/TGRS.2010.2049023, 2011.

---

## Author Response (AR2)

**"Deriving surface soil moisture from reflected GNSS signal observations from a grassland site in southwestern France"**
**by Sibo Zhang et al.**

**Cover letter to the editor**

20 February 2018

Dear Dr. Miriam Coenders-Gerrits,

Pleas find enclosed a marked-up version of the revised manuscript accounting for minor changes in response to the last comments from Reviewer 1:

- Pg 1, Line 9: I would say "This work assesses" instead of "This work aims to assess."
- Pg 1, Line 25: Capitalize the first letters of the words in the acronyms SMAP and SMOS.
- Pg 1, Line 26: Replace 'in' with 'of' in "These products consist in surface…"
- Pg 2, Line 2: Replace "In particular, developing new…" with "In particular, the development of new…"
- Pg 6, Line 13: Replace "markedly increase" with "markedly increases"

**RESPONSE**: Yes, these editorial changes were made.

- Pg 7, first paragraph: I think this paragraph can be removed or shortened, since you are no longer comparing the two algorithms.

**RESPONSE**: We think the whole paragraph is needed in order to define the S parameter discussed in Section 4.2 ("results very similar to those presented in Fig. 5 can be obtained by multiplying by 0.6 the S value 25 used by Chew et al. (2016)").

- Pg 10, 2nd paragraph: By splitting up the time series, are you still using the observed soil moisture max/min values for the entire time series, or for each segment?

**RESPONSE**: Yes, we added the following sentence: "The observed soil moisture minimum and maximum values are derived for each time segment".

- And finally, I would request that the authors consider whether or not the number of tables and figures is necessary to present their results. I worry that with so many tables, readers will not take the time to ingest the most important parts of the work and might get lost in the details.

**RESPONSE**: Yes, we merged former Tables 2, and 6 in a single Table 2. We deleted former Table 5.

Yours sincerely,
Jean-Christophe Calvet, Sibo Zhang.

[revised manuscript text omitted]

5 $N$ is the number of observations.

[revised manuscript text omitted]

**by Sibo Zhang et al.**

**Cover letter to the editor**

5 February 2018

Dear Dr. Miriam Coenders-Gerrits,

The authors' response to the comments of the two anonymous referees has been published on the HESS web site.

In response to comments by Reviewer 1, we removed the comparison with Chew et al. (2016) in the text, in Tables 3 and 4, and in Figs. 4, 5, 6. We also explained in the Discussion section how VSM retrieval in marked changing vegetation conditions could be performed without using ancillary in situ VSM observations. Such changes in vegetation conditions can be identified from the GNSS signal and then the retrieved VSM values can be rescaled to be consistent with a land surface model, using a data assimilation framework. Our results show that using this rescaling technique would be feasible since the ISBA simulations of VSM correlate well with the retrieved VSM (Fig. 8). The main reason for this result is that ISBA is forced by the SAFRAN atmospheric analysis, incorporating a large number of in situ raingauge observations. This is another way of using ancillary in situ observations.

All changes relative to the published HESS paper are detailed in the marked-up version of the new manuscript. They include all the response elements given by the authors in response to the reviewers' comments (green and blue for Reviewer 1 and 2, respectively). Other changes in the text are in red.

In response to your comment, we have elaborated more on the rescaling issue (in yellow) and added other published examples of the assimilation of satellite-derived VSM observations making use of the cumulative distribution functions (CDF) matching to rescale observations prior to their assimilation in a land surface model (Reichle and Koster, 2004; Draper and Reichle, 2015).

References:

Draper, C. and Reichle, R.: The impact of near-surface soil moisture assimilation at subseasonal, seasonal, and inter-annual timescales, Hydrol. Earth Syst. Sci., 19, 4831–4844, https://doi.org/10.5194/hess-19-4831-2015, 2015.

Reichle, R. and Koster, R.: Bias reduction in short records of satellite soil moisture, Geophys. Res. Lett., 31, L19501, https://doi.org/10.1029/2004GL020938, 2004.

Yours sincerely,

Jean-Christophe Calvet, Sibo Zhang.

**LIST OF CHANGES MADE IN RESPONSE TO COMMENTS OF REVIEWER #1**

**1.1 [General comments: The authors utilize a geodetic-quality GNSS antenna (AR10 type) in a meadow to test out a soil moisture retrieval algorithm under different stages of natural grass cover growth. They find that their retrieval algorithm performs well and retrieves soil moisture compared to in situ with an RMSE less than 0.04 cm3 cm-3. They compare their results to a 'benchmark' algorithm and find that their algorithm performs better. They also vary the height of the antenna to see if antenna height affects their results, and they also look at the effects from changing the sampling rate. They find that antenna height does not affect their retrievals, but sampling rate does. Overall, there are two major short comings of this study that must be addressed: \*First, the 'benchmark' algorithm that the authors compare their own retrievals to should NOT be used for this type of antenna. The benchmark algorithm developed in Chew et al. (2016) was created solely for the antennas used in the Plate Boundary Observatory network (Trimble antennas). It is well known that the algorithm would need to be calibrated for use with a different antenna type. The authors should remove the portion of the paper (figures and text) that compare their algorithm to that from Chew et al. (2016). This is a significant portion of the text and discussion that should be removed, but the paper is still worthwhile without it.]**

**Response 1.1:** changes in the marked-up version of the new manuscript

**P. 7, L. 5-13:**
"Due to the good linear relationship between $\phi$ and in situ surface VSM, VSM can be estimated for each satellite track (Chew et al., 2016):

$$VSM = S \cdot (\phi - \phi_{\min}) + VSM_{resid} \tag{2}$$

The $S$ parameter (in $m^3$ $m^{-3}$ degree$^{-1}$) is defined using the a priori value. A value of $S = 0.0148$ $m^3$ $m^{-3}$ degree$^{-1}$ was proposed by Chew et al. (2016) for the PBO-H$_2$O network. This value is adapted to situations of low vegetation density or cover and is valid for the Trimble antennas used in the PBO-H$_2$O network. In this equation, the $\phi$ time series is zeroed using a minimum phase value ($\phi_{\min}$) for each satellite track. This procedure is useful to ensure compatibility among different satellite tracks. $\phi_{\min}$ is the mean of the lowest 15% of $\phi$ values for each satellite track during the considered time segment and VSM$_{resid}$ is the residual (minimum) volumetric soil moisture value."

**P. 7, L. 15:**
"In this study, the method proposed by Zhang et al. (2017) is used."

**P. 10, L. 20-21:**
"It is interesting to note that results very similar to those presented in Fig. 5 can be obtained by multiplying by 0.6 the S value used by Chew et al. (2016) (not shown)."

**1.2 [\*Second, the fact that the authors' retrieval algorithm requires having in situ observations of maximum and minimum soil moisture (Eq. 3) detracts significantly from the usefulness of the algorithm. Of course their algorithm produces soil moisture retrievals within the bounds of the in situ probes–it is effectively scaled by the in situ**

observations. **Furthermore, the authors state that they need min/max in situ observations from both vegetation growth and senescence periods, which then means that they need ancillary vegetation information in order for their algorithm to work. If you need vegetation data and in situ soil moisture probes in order for your algorithm to work, why use GNSS-IR at all? The authors should spend some time re-working their algorithm so that they don't need in situ soil moisture information. If the authors can address the above two comments, then the paper will be technically correct and will make a more worthwhile contribution to the field of GNSS-IR in general. I know that these are harsh criticisms, and I don't want the authors/editors to think that I don't like the paper–overall, I enjoyed reading it. It is well organized and clearly written. I think reporting their retrieval results is worthwhile, and removing the comparison with the benchmark algorithm will not detract from the paper.]**

**Response 1.2:** changes in the marked-up version of the new manuscript

**P. 12, L. 4-16:**
"Figure 7 clearly shows that using GNSS-IR to retrieve VSM values in $m^3$ $m^{-3}$ when significant changes in vegetation effects occur is challenging. The need to harmonize VSM retrievals from TS3 and TS4 is related to the cutting of the grass when vegetation effects are pronounced ($A_{norm}$ is lower than 0.78, see Fig. 1). As a consequence, monitoring VSM using a GNSS network could be difficult when vegetation effects are noticeable. However, we show that one may use the information from $A_{norm}$ data to define time segments for which scaled VSM time series are valid. For example, grass cutting can be detected from the rapid rise in $A_{norm}$ value. In this study, we used independent VSM in situ observations to harmonize the VSM time series across TS3 and TS4. Since in situ observations are not extensively available, this technique is not readily applicable at other sites. In practice, one could possibly use a data assimilation framework able to integrate the VSM retrievals into model VSM simulations such as those produced by the ISBA land surface model (Albergel et al., 2017). In such Land Data Assimilation Systems (LDAS), a complex seasonal rescaling of VSM observations is needed (Reichle and Koster, 2004; Draper and Reichle, 2015), especially when the observations are not properly decontaminated from vegetation effects (Stoffelen et al., 2017)."

**P. 15, L. 19-24:**
"More experiments over contrasting vegetation types are needed to further examine the feasibility of integrating GNSS-IR retrievals in land surface models. Land data assimilation systems are usually used for satellite observations but can also integrate ground observations. In such a framework, model simulations of vegetation biomass and soil moisture could be combined with GNSS-IR retrievals. Proposing a complete protocol to apply this method to local GNSS antennas would require observations at a large number of sites. More research is needed to use GNSS-IR in densely vegetated areas."

New references:
**P. 16, L. 1-5:**
"Albergel, C., S. Munier, D. J. Leroux, H. Dewaele, D. Fairbairn, A. L. Barbu, E. Gelati, W. Dorigo, S. Faroux, C. Meurey, P. Le Moigne, B. Decharme, J.-F. Mahfouf, J.-C. Calvet: Sequential assimilation of satellite-derived vegetation and soil moisture products using SURFEX_v8.0: LDAS-Monde assessment over the Euro-Mediterranean area, Geosci. Model Dev., Geosci. Model Dev., 10, 3889–3912, https://doi.org/10.5194/gmd-10-3889-2017, 2017."

**P. 19, L. 19-21:**
"Stoffelen, A., S. Aaboe, J.-C. Calvet, J. Cotton, G. De Chiara, J. Figua-Saldana, A. A. Mouche, M. Portabella, K. Scipal, W. Wagner: Scientific developments and the EPS-SG scatterometer, IEEE J. Sel. Topics Appl. Earth Obs. Remote Sens., 10 (5), 2086-2097, https://doi.org/10.1109/JSTARS.2017.2696424, 2017."

**1.3 [Specific comments: Page 2, line 5: You should make it clear that GNSS-IR is not used for spaceborne applications, as you reference in the Camps et al. (2008) paper. The spaceborne technique is very different from GNSS-IR.]**

**Response 1.3:** changes in the marked-up version of the new manuscript

**P. 2, L. 4-5:**
This sentence was deleted.

**1.4 [Page 8, line 9: Isn't another way of saying this, is that the sensing depth of GNSS-IR is less than 5 cm? This has been found in previous studies for GNSS-IR (Chew et al., 2014) and for L-band microwave remote sensing in general (Shellito et al., 2016, GRL). The comparison with the land surface model is a bit rushed and perhaps not needed. As you know, there are a variety of different land surface models, each with their own parameterizations of the land surface. There aren't enough details provided about the land surface model for readers to understand its advantages and shortcomings. Was it parameterized for this particular field? What is the spatial resolution of the model? The authors do not spend much time with comparing their results to the model output, so it would be easy to remove this part of the paper.]**

**Response 1.4:** changes in the marked-up version of the new manuscript

**P. 5, L. 8-16:**
"The ISBA model used the atmospheric forcing data produced by the SAFRAN atmospheric analysis of Météo-France. The model version used in this study was designed for generic country-scale simulations over France at a spatial resolution of 8 km x 8 km. It was not calibrated for this particular site. Sub-grid vegetation types are represented and soil moisture and soil temperature profiles are simulated for each vegetation type, independently of other vegetation types. In this study, the C3 grassland plant functioning type and a multilayer representation of the soil hydrology are considered. The model soil depth is 12 m, with 15 layers and the layer thickness increases from the top surface layer to the deepest layers (Decharme et al., 2011). It must be noted that the SAFRAN precipitation forcing is based on ground observations and is quite realistic (Quintana-Segui et al., 2008)."

**P. 9, L. 23-25:**
"This difference reduces the correlation and increases the errors and can be attributed to a GNSS-IR sensing depth less than 5 cm (Chew et al., 2014 ; Shellito et al., 2016), in relation to vegetation litter effects (see Sect. 5.3)."

New references:
**P. 16, L. 23-25:**

"Decharme, B., Boone, A., Delire, C., and Noilhan, J.: Local evaluation of the Interaction between Soil Biosphere Atmosphere soil multilayer diffusion scheme using four pedotransfer functions, J. Geophys. Res., 116, D20126, https://doi.org/10.1029/2011JD016002, 2011."

**P. 18, L. 16-18:**

"Quintana-Segui, P., Lemoigne, P., Durand, Y., Martin, E., Habets, F., Baillon, M., Canellas, C., Franchisteguy, L., and Morel, S.: Analysis of near surface atmospheric variables: Validation of the SAFRAN analysis over France, J. Appl. Meteorol. Clim., 47, 92–107, https://doi.org/10.1175/2007JAMC1636.1, 2008."

**P. 19, L. 10-13:**

"Shellito, P. J., Small, E. E., Colliander, A., Bindlish, R., Cosh, M. H., Berg, A. A., Bosch, D. D., Caldwell, T. G., Goodrich, D. C., McNairn, H., Prueger, J. H., Starks, P. J., van der Velde, R., and Walker, J. P.: SMAP soil moisture drying more rapid than observed in situ following rainfall events, Geophys. Res. Lett., 43, 8068–8075, https://doi.org/10.1002/2016GL069946, 2016."

**1.5 [With regards to the sampling rate discussion–are you not just exploring effects of sampling lower than the required Nyquist sampling frequency for a given antenna height?]**

**Response 1.5:** changes in the marked-up version of the new manuscript

**P. 14, L. 8-9:**

"It should be noted that in the examples illustrated by Figs. 9, 10, and 11 the SNR frequency is always lower than the Nyquist frequency."

**1.6 [Technical corrections: Figure 2 needs a second y-axis for Anorm. I understand they are scaled between 0-1 just like you have your biomass values, but it's a bit confusing without an extra label. ]**

**Response 1.6:** changes in the marked-up version of the new manuscript

**P. 30:**

A second y-axis was added for $A_{norm}$.

**LIST OF CHANGES MADE IN RESPONSE TO COMMENTS OF REVIEWER #2**

**2.1 [OVERVIEW**
**The manuscript investigates the use of the Global Navigation Satellite System Interferometric Reflectometry (GNSS-IR) technique for soil moisture retrieval. Specifically, one year of observations were acquired at a grassland site in France by using an antenna at 2 different heights (3.3 and 29.4 m). GNSS-IR data are compared with ground-based reference measurements and the effect of vegetation, litter water interception, sampling interval and antenna height is analysed on the accuracy of the measurements.**
**GENERALCOMMENTS**
**GNSS-IR represents a new approach for measuring soil moisture and surely deserves to be investigated. Specifically, the potential of using GNSS-IR measurements for monitoring soil moisture over large areas might represent an important step forward in our capability of measuring soil moisture at field scale. The manuscript is well written and clear and, hence, I have no major comments to be addressed. I believe the paper might be published after considering the minor comment I reported below.**
**1) The same authors (nearly) published a paper in 2017 with a very similar purpose. I can see the differences between the two papers, and hence I believe this paper should be published. However, I strongly suggest to clearly underline the differences between the two papers and the main innovative aspects (e.g., antenna height, analysis of vegetation effect) of the current study.]**

**Response 2.1:** changes in the marked-up version of the new manuscript

The differences between the two studies is now better described throughout the manuscript.

**P. 3, L. 12-14:**
"Zhang et al. (2017) used the GNSS-IR technique for a wheat field throughout the growth and senescence period in 2015. The L1 C/A signal was acquired over a wheat field during a period of about 7 months using a Leica GR25 receiver, and a Leica AR10 antenna at a constant height of 2.5 m above the soil surface."

**P. 3, L. 17-19:**
"In this study, both L2C and L5 signals were acquired over a meadow during a rather long period of time of about 15 months using the same equipment (GR25 receiver, AR10 antenna) at contrasting heights (3.3 or 29.4 m) above the soil surface."

**P. 3, L. 26-29:**
"A key difference between this study and Zhang et al. (2017) is related to the type of observed vegetation canopy. The meadow considered in this study and the wheat field considered by Zhang et al. (2017) present contrasting characteristics. The meadow is cut once a year and consists of a multi-species permanent grassland incorporating a litter composed of dead leaves. On the other hand, the wheat crop in Zhang et al. (2017) consisted of a single plant species with no litter."

**P. 4, L. 25-27:**
"The grass height did not exceed 0.3 m during the experiment time period. This is much lower than maximum height of the wheat crop (~ 1 m) in Zhang et al. (2017). A large difference

could also be noticed in maximum above-ground dry biomass values: less than 0.5 kg m$^{-2}$ for grass (this study), about 1 kg m$^{-2}$ for wheat (Zhang et al., 2017)."

**P. 11, L. 12-15:**
"While VSM could not be retrieved by Zhang et al. (2017) after wheat tillering, i.e. for plant height larger than 0.2 m, we could retrieve scaled VSM values throughout time segments of the grass growing and senescence phases. However, retrieving VSM values in m$^3$ m$^{-3}$ was challenging and required a seasonal rescaling to account for vegetation effects (see Fig. 7)."

**2.2 [2) In the description of the study area, more details should be provided. At which depth are installed the surface measurements? How many rain gauges are available in the study area? How many soil moisture stations (I guess one)? Which model is used for simulating soil moisture? I suggest adding all these details in the revised manuscript.]**

**Response 2.2:** changes in the marked-up version of the new manuscript

**P. 4, L. 29 - P. 5, L. 2:**
"Mean in situ VSM observations at 5 and 1 cm depths were performed using precise Delta-T ML2x ThetaProbes and low-cost Decagon EC-5 VSM sensors, respectively. Three ThetaProbes measured VSM at a depth of 5 cm and were located within a few meters of each other (red star in Fig. 1). The mean value was derived from these probes to represent the in situ VSM observations at 5 cm. Only one EC-5 sensor was used to measure VSM at 1 cm. Precipitation measurements were made in the experimental field by one rain gauge close to the in situ soil moisture sensors."

**P. 5, L. 6-9:**
"VSM simulations for the top 1 cm were produced using the ISBA (Interactions between Soil, Biosphere, and Atmosphere) land surface model within the SURFEX (version 8.0) modeling platform (Masson et al., 2013). In addition to VSM, simulations included the soil iced water content and the vegetation above-ground dry biomass. The ISBA model used the atmospheric forcing data produced by the SAFRAN atmospheric analysis of Météo-France."

**2.3 [3) At Page 5, lines 17-20 it reads that only some satellite tracks are selected based on the comparison with in situ measurements. I was wondering how the authors will select the tracks if in situ soil moisture observations are not available, it should be clarified.]**

**Response 2.3:** changes in the marked-up version of the new manuscript

**P. 6, L. 10-15:**
"The selection of satellite tracks and elevation angles was performed by comparing VSM retrievals with the in situ VSM observations described in Sect. 2.1. It must be noted that this limitation only affected measurements at a height of 29.4 m and was caused by the more complex experimental constraints in this configuration (e.g. possible parasitic signal reflection on buildings). For the low antenna configuration (3.3 m), this additional data sorting was not needed and all available satellite tracks with a complete elevation angle range (between 7 and 30°) were used. As a result, a larger variety of satellite tracks could be used for the antenna at a height of 3.3 m with 1 s sampling."

**2.4 [4) At Page 5, line 28, what is the "multipath interference pattern". Please clarify.]**

**Response 2.4:** changes in the marked-up version of the new manuscript

**P. 6, L. 20-27:**
"For a static receiver, the SNR is governed to a large extent by the interference pattern (IP). The IP is defined as the coherent summation of direct and reflected GNSS signals on the in-phase and quadrature space (Zavorotny et al., 2014). This coherent summation generates an IP where high and intermediate frequencies distinct from noise frequencies, are related to the difference of travelled distance between direct and reflected waves. The IP can be characterized with GNSS receivers using either (1) two antennas (e.g. Rodriguez-Alvarez et al., 2011) or (2) one antenna (e.g. Larson et al., 2008; Chew et al., 2014; Zhang et al., 2017). In this study we used the one-antenna IP technique as illustrated by Fig. 1 in Larson et al. (2016) for a simple planar and horizontal ground reflection."

New reference:
**P. 18, L. 21-24:**

[revised manuscript text omitted]